# MIRAGE: MODEL-AGNOSTIC GRAPH DISTILLATION FOR GRAPH CLASSIFICATION

**Mridul Gupta***
Yardi School of Artificial Intelligence
Indian Institute of Technology, Delhi
Hauz Khas, New Delhi, Delhi, India
mridul.gupta@scai.iitd.ac.in

**Sahil Manchanda***
Department of Computer Science
Indian Institute of Technology, Delhi
Hauz Khas, New Delhi, Delhi, India
sahilm1992@gmail.com

**Hariprasad Kodamana**
Department of Chemical Engineering
Yardi School of Artificial Intelligence
Indian Institute of Technology, Delhi
Hauz Khas, New Delhi, Delhi, India
kodamana@iitd.ac.in

**Sayan Ranu**
Department of Computer Science
Yardi School of Artificial Intelligence
Indian Institute of Technology, Delhi
Hauz Khas, New Delhi, Delhi, India
sayanranu@cse.iitd.ac.in

## ABSTRACT

GNNs, like other deep learning models, are data and computation hungry. There is a pressing need to scale training of GNNs on large datasets to enable their usage on low-resource environments. Graph distillation is an effort in that direction with the aim to construct a smaller synthetic training set from the original training data without significantly compromising model performance. While initial efforts are promising, this work is motivated by two key observations: (1) Existing graph distillation algorithms themselves rely on training with the full dataset, which undermines the very premise of graph distillation. (2) The distillation process is specific to the target GNN architecture and hyper-parameters and thus not robust to changes in the modeling pipeline. We circumvent these limitations by designing a distillation algorithm called MIRAGE for graph classification. MIRAGE is built on the insight that a message-passing GNN decomposes the input graph into a *multiset* of *computation trees*. Furthermore, the frequency distribution of computation trees is often skewed in nature, enabling us to condense this data into a concise distilled summary. By compressing the computation data itself, as opposed to emulating gradient flows on the original training set—a prevalent approach to date—MIRAGE transforms into an architecture-agnostic distillation algorithm. Extensive benchmarking on real-world datasets underscores MIRAGE's superiority, showcasing enhanced generalization accuracy, data compression, and distillation efficiency when compared to state-of-the-art baselines.

## 1 INTRODUCTION AND RELATED WORK

GNNs have shown state-of-the-art performance in various machine learning tasks, including node classification (Hamilton et al., 2017; Veličković et al., 2018), graph classification (Ying et al., 2021; Rampášek et al., 2022), and graph generative modeling (Vignac et al., 2023; You et al., 2018; Goyal et al., 2020; Gupta et al., 2022; Manchanda et al., 2023). Their applications percolate various domains including social networks (Manchanda et al., 2020; Chakraborty et al., 2023; Wang et al., 2021; Yang et al., 2011), traffic forecasting (Gupta et al., 2023; Jain et al., 2021; Li et al., 2020; Wu et al., 2017), modeling of physical systems (Bishnoi et al., 2023; Bhattoo et al., 2022; Sanchez-Gonzalez et al., 2020) and several others. Despite the efficacy of GNNs, like many other deep-learning models, GNNs are data, as well as, computation hungry. One important area of study that tackles this problem is the idea of *data distillation* (or condensation) for graphs. Data distillation seeks to compress the vital information within a graph dataset while preserving its critical structural and functional properties. The objective in the distillation process is to compress the train data as much as possible without compromising on the predictive accuracy of the GNN when trained on the distilled data. The distilled data therefore significantly alleviates the computational and storage

---

*Denotes Equal Contribution

demands, due to which GNNs may be trained more efficiently including on devices with limited resources, like small chips. It is important to note that the distilled dataset need not be a subset of the original data; it may be a fully synthetic dataset.

## 1.1 EXISTING WORKS

Data distillation has proven to be an effective strategy for alleviating the computational demands imposed by deep learning models. For instance, in the case of DC (Zhao et al., 2021), a dataset of $\approx 60,000$ images was distilled down to just $100$ images, resulting in an impressive accuracy of $97.4\%$, compared to the original accuracy of $99.6\%$.

Graph distillation has also been explored in prior research (Jin et al., 2022; 2021; Xu et al., 2023). These graph distillation algorithms share a common approach, where the distilled dataset seeks to replicate the same gradient trajectory of the model parameters as seen in the original training set. In this work, we observe that the process of mimicking gradients necessitates supervision from the original training set, giving rise to significant limitations.

1. **Counter-objective design:** The primary goal in data distillation is to circumvent the need for training on the entire training dataset, given the evident computational and storage constraints. Paradoxically, existing algorithms aim to replicate the gradient trajectory of the original dataset, necessitating training on the full dataset for distillation. Consequently, the fundamental premise of data distillation is compromised.

2. **Dependency on Model and Hyper-Parameters:** The gradients of model weights are contingent on various factors such as the specific GNN architecture and hyper-parameters, including the number of layers, hidden dimensions, dropout rates, and more. As a result, any alteration in the architecture, such as transitioning from a Graph Convolutional Network (GCN) to a Graph Attention Network (GAT), or adjustments to hyper-parameters, necessitates a fresh round of distillation. It has been shown in the literature (Yang et al., 2023), and also substantiated in our empirical study (Appendix C), that there is a noticeable drop in performance if the GNN architecture used for distillation is different from the one used for eventual training and inference.

3. **Storage Overhead:** Given the dependence of the distillation process on both the GNN architecture and hyper-parameters, a distinct distilled dataset must be maintained for each unique combination of architecture and hyper-parameters. This inevitably amplifies the storage requirements and maintenance overhead.

## 1.2 CONTRIBUTIONS

To address the above outlined limitations of existing algorithms, we design a graph distillation algorithm called MIRAGE for graph classification. MIRAGE proposes several innovative strategies imparting significant advantages over existing graph distillation methods.

- **Model-agnostic algorithm:** Instead of replicating the gradient trajectory, MIRAGE emulates the input data processed by message-passing GNNs[1]. By shifting the computation task to the pre-learning phase, MIRAGE and the resulting distilled data become independent of hyper-parameters and model architecture (as long as it adheres to a message-passing GNN framework like GAT (Veličković et al., 2018), GCN (Kipf & Welling, 2016), GRAPHSAGE (Hamilton et al., 2017), GIN (Xu et al., 2019), etc.). Moreover, this addresses a critical limitation of existing graph distillation algorithms that necessitate training on the entire dataset.

- **Novel GNN-customized algorithm:** MIRAGE exploits the insight that given a graph, an $\ell$-layered message-passing GNNs decomposes the graph into a set of computation trees of depth $\ell$. Furthermore, the frequency distribution of computation trees often follows a power-law distribution (See. Fig. 2). MIRAGE exploits this pattern by mining the set of frequently co-occurring trees. Subsequently, the GNN is trained by sampling from the co-occurring trees. An additional benefit of this straightforward distillation process is its computational efficiency, as the entire algorithm can be executed on a CPU. This stands in contrast to existing graph distillation algorithms that rely on GPUs, making MIRAGE a more resource and environment friendly alternative.

- **Empirical performance:** We perform extensive benchmarking of MIRAGE against state-of-the-art graph distillation algorithms on six real world-graph datasets and establish that MIRAGE achieves *(1)* higher prediction accuracy on average, *(2)* 4 to 5 times higher data compression, and *(3)* a significant 150-fold acceleration in the distillation process when compared to state-of-the-art graph distillation algorithms.

---

[1]Hence, the name MIRAGE.

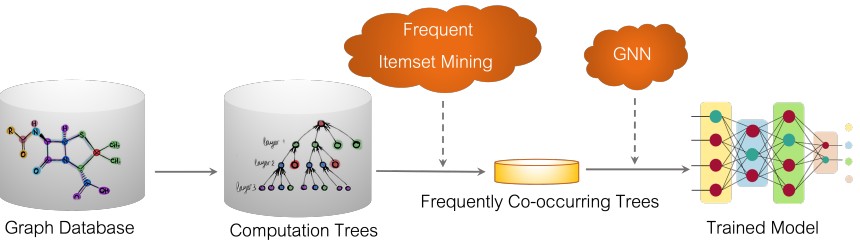

Figure 1: Pipeline of MIRAGE.

## 2 PRELIMINARIES AND PROBLEM FORMULATION

**Definition 1** (Graph). *A graph is defined as $\mathcal{G} = (\mathcal{V}, \mathcal{E}, \boldsymbol{X})$ over a finite non-empty node set $\mathcal{V}$ and edge set and $\mathcal{E} = \{(u, v) \mid u, v \in \mathcal{V}\}$. $\boldsymbol{X} \in \mathbb{R}^{|\mathcal{V}| \times |F|}$ is a node feature matrix where $F$ is a set of features characterizing each node.*

As an example, in case of molecules, nodes and edges would correspond to atoms and bonds, respectively, while features would correspond to properties such as atom type, hybridisation state, etc.

Equivalence between graphs is captured through *graph isomorphism*.

**Definition 2** (Graph Isomorphism). *Two graphs $\mathcal{G}_1$ and $\mathcal{G}_2$ are considered isomorphic (denoted as $\mathcal{G}_1 \cong \mathcal{G}_2$) if there exists a bijection between their node sets that preserves the edges and node features. Specifically, $\mathcal{G}_1 \cong \mathcal{G}_2 \iff \exists f : \mathcal{V}_1 \to \mathcal{V}_2$ such that: (1) $f$ is a bijection, (2) $\mathbf{x}_v = \mathbf{x}_{f(v)}$,[2] and (3) $(u, v) \in \mathcal{E}_1$ if and only if $(f(u), f(v)) \in \mathcal{E}_2$.*

**Graph Classification:** In graph classification, we are given a set of *train* graphs $\mathcal{D}_{tr} = \{\mathcal{G}_1, \cdots, \mathcal{G}_m\}$, where each graph $\mathcal{G}_i$ is tagged with a class label $\mathcal{Y}_i$. The objective is to train a GNN $\Phi_{\Theta_{tr}}$ parameterized by $\Theta_{tr}$ from this train set such that given an unseen set of *validation* graphs $\mathcal{D}_{val}$ with unknown labels, the label prediction error is minimized. Mathematically, this involves learning the optimal parameter set $\Theta_{tr}$, where:

$$\Theta_{tr} = \arg \min_{\Theta} \left\{ \epsilon \left( \{ \Phi_{\Theta} \left( \mathcal{G} \right) \mid \forall \mathcal{G} \in \mathcal{D}_{val} \} \right) \right\} \tag{1}$$

Here, $\Phi_{\Theta_{tr}}(\mathcal{G})$ denotes the predicted label of $\mathcal{G}$ by GNN $\Phi_{\Theta_{tr}}$ and $\epsilon \left( \{ \Phi_{\Theta} \left( \mathcal{G} \right) \mid \forall \mathcal{G} \in \mathcal{D}_{val} \} \right)$ denotes the *error* with parameter set $\Theta$. Error may be measured using any of the known metrics such as cross-entropy loss, negative log-likelihood, etc.

Hereon, we implicitly assume $\Phi$ to be a *message-passing* GNN (Kipf & Welling, 2016; Hamilton et al., 2017; Veličković et al., 2018; Xu et al., 2019). Furthermore, we assume the validation set to be fixed. Hence, the generalization error of GNN $\Phi$ when trained on dataset $\mathcal{D}_{tr}$ is simply denoted using $\epsilon_{\mathcal{D}_{tr}}$. The problem of graph distillation for graph classification is now defined as follows.

**Problem 1** (Graph Distillation). *Given a training set and validation set of graphs, $\mathcal{D}_{tr}$ and $\mathcal{D}_{val}$, respectively, generate a dataset $\mathcal{S}$ from $\mathcal{D}_{tr}$ with the following dual objectives:*

1. **Error:** *Minimize the error gap between $\mathcal{S}$ and $\mathcal{D}_{tr}$ on the validation set, i.e., minimize $\{|\epsilon_{\mathcal{S}} - \epsilon_{\mathcal{D}_{tr}}|\}$.*
2. **Compression:** *Minimize the size of $\mathcal{S}$. Size will be measured in terms of raw memory consumption, i.e., in bytes.*

In addition to the above objectives, we impose two practical constraints on the distillation algorithm. First, it should not rely on the specific GNN architecture, except for the assumption that it belongs to the message-passing family. Second, it should be independent of the model parameters when trained on the original training set. Adhering to these constraints addresses the limitations outlined in § 1.1.

## 3 MIRAGE: PROPOSED METHODOLOGY

MIRAGE exploits the computation framework of message-passing GNNs to craft an effective data compression strategy. Fig. 1 presents the pipeline of MIRAGE. GNNs decompose any graph into a collection of *computation trees*. In Fig. 2, we plot the frequency distribution of computation trees across various graph datasets. We observe that the frequency distribution follows a power-law. This distribution indicates that a compact set of top-$k$ frequent trees effectively captures a substantial portion of the distribution mass while retaining a wealth of information content. Empowered with this observation, in MIRAGE, the GNN is trained only through the frequent tree sets. We next elaborate on each of these intermediate steps.

[2]One may relax feature equivalence to having a distance within a certain threshold.

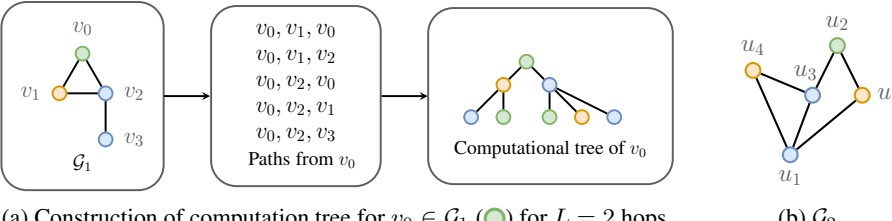

(a) Construction of computation tree for $v_0 \in \mathcal{G}_1$ (⬤) for $L = 2$ hops.          (b) $\mathcal{G}_2$.

Figure 3: In (a) we show the construction of the computation tree for $v_0 \in \mathcal{G}_1$. In (b), we present $\mathcal{G}_2$, which has an isomorphic 2-hop computational tree for $u_2$ despite its neighborhood being non-isomorphic to $v_0$. We assume the node feature vectors to be an one-hot encoding of the node colors.

### 3.1 COMPUTATION FRAMEWORK OF GNNS

GNNs aggregate messages in a layer-by-layer manner. Assuming $\mathbf{x}_v \in \mathbb{R}^{|F|}$ as the input feature vector for every node $v \in \mathcal{V}$, the $0^{th}$ layer representation of node $v$ is simply $\mathbf{h}_v^0 = \mathbf{x}_v \ \forall v \in \mathcal{V}$. Subsequently, in each layer $\ell$, GNNs draw messages from its neighbours $\mathcal{N}_v^1$ and aggregate them as follows:

$$\mathbf{m}_v^\ell(u) = \text{MSG}^\ell(\mathbf{h}_u^{\ell-1}, \mathbf{h}_v^{\ell-1}) \ \forall u \ \in \mathcal{N}_v^1 \qquad (2)$$

$$\overline{\mathbf{m}}_v^\ell = \text{AGGREGATE}^l(\{\!\{\mathbf{m}_v^\ell(u), \forall u \in \mathcal{N}_v\}\!\}) \qquad (3)$$

where $\text{MSG}^\ell$ and $\text{AGGREGATE}^\ell$ are either pre-defined functions (Ex: MEANPOOL) or neural networks (GAT (Veličković et al., 2018)). $\{\!\{\cdot\}\!\}$ denotes a multi-set since the same message may be received from multiple nodes. The $\ell^{th}$ layer representation of $v$ is a summary of all the messages drawn.

$$\mathbf{h}_v^\ell = \text{COMBINE}^\ell(\mathbf{h}_v^{\ell-1}, \overline{\mathbf{m}}_v^\ell) \qquad (4)$$

where $\text{COMBINE}^\ell$ is a neural network. Finally, the representation of the graph is computed as:

$$\mathbf{h}_\mathcal{G} = \text{COMBINE}(\mathbf{h}_v^L, \forall v \in \mathcal{V}) \qquad (5)$$

Here, COMBINE could be aggregation functions such as MEANPOOL, SUMPOOL, etc. and $L$ is total number of layers in the GNN.

### 3.2 COMPUTATION TREES

We now define the concept of *computation trees* and draw attention to some important properties that sets the base for graph distillation.

**Definition 3** (Computation Tree). *Given graph $\mathcal{G}$, node $v$ and the number of layers $L$ in a GNN, we construct a computation tree $\mathcal{T}_v^L$ rooted at $v$. Starting from $v$, enumerate all paths, including non-simple paths [3], of $L$ hops. Next, merge these paths under the following constraints to form $\mathcal{T}_v^L$. Two nodes $v_i$ and $v_j'$ in paths $P = \{v_0 = v, v_1, \cdots, v_L\}$ and $P' = \{v_0' = v, v_1', \cdots, v_L'\}$, respectively, are merged into a single node in $\mathcal{T}_v^L$ if either $i = j = 0$ or $v_i = v_j'$, $i = j$ and $\forall k \in [0, i-1]$, $v_k$ and $v_k'$ have been merged.*

**Observation 1.** *In an $L$-layered GNN, the final representation $\mathbf{h}_v^L$ of a node $v$ in graph $\mathcal{G}$ can be computed from its computation tree $\mathcal{T}_v^L$.*

*Proof.* In each layer, a GNN draws messages from its direct neighbors. Over $L$ layers, a node $v$ receives messages from nodes reachable within $L$ hops. All paths of length up to $L$ from $v$ are contained within $\mathcal{T}_v^L$, Hence, the computation tree is sufficient for computing $\mathbf{h}_v^L$. ☐

**Observation 2.** *If $\mathcal{T}_v^L \cong \mathcal{T}_u^L$, then $\mathbf{h}_v^L = \mathbf{h}_u^L$.*

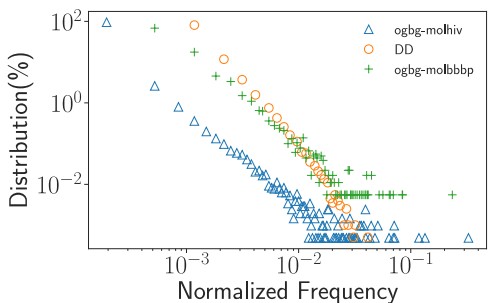

Figure 2: Frequency distribution of computation trees across datasets. The "frequency" of a computation tree denotes the number of occurrences of that specific tree across all graphs in a dataset. The *normalized* frequency of a tree is computed by dividing its frequency with the total number of graphs in at dataset and thus falls in the range $[0, 1]$. The $x$-axis of the plot depicts the normalized frequency counts observed in a dataset, while the $y$-axis represents the percentage of computation trees corresponding to each frequency count. Both $x$ and $y$ axes are in log scale. The distribution is highly skewed characterized by a dominance of trees with low frequency counts, while a small subset of trees exhibiting higher frequencies. For example, in ogbg-molhiv, the most frequent tree alone has normalized frequency of $0.32$.

---

[3]a non-simple path allows repetition of vertices

*Proof.* A message-passing GNN is at most as powerful as *Weisfeiler-Lehman tests (1-WL)* (Xu et al., 2019), which implies that if the $L$-hop neighborhoods of nodes $u$ and $v$ are indistinguishable by 1-WL, then their representations would be the same. 1-WL cannot distinguish between graphs of identical computation trees (Shervashidze et al., 2011). □

**Observation 3.** *Two nodes with non-isomorphic $L$-hop neighborhoods may have isomorphic computation trees.*

PROOF. See Figure. 3. □

**Implications:** Obs. 1 reveals that any graph may be decomposed into a *multiset* of computation trees (not a set since the same tree may appear multiple times) without loosing any information. By learning the representations of each computation tree root, we can construct each node representation accurately, and consequently, derive an accurate representation for the entire graph (Recall Eq. 5). Now, suppose the frequency distribution of these computation trees in the multiset is significantly skewed, with a small minority dominating the count. In that case, the graph representation, obtained by aggregating the root representations of only the highly frequent trees, will closely approximate the true graph representation. This phenomenon, illustrated in Figure. 2, is commonly observed. Furthermore, Obs. 3 implies that the set of all computations trees is strictly a subset of the set of all $L$-hop subgraphs in the dataset, leading to further skewness in the distribution. Leveraging this pattern, we devise a distillation process that revolves around retaining only those computation trees that *co-occur* frequently. While frequency captures the contribution of a computation tree towards the graph representation, co-occurrence among trees captures frequent graph compositions.

## 3.3 MINING FREQUENTLY CO-OCCURRING COMPUTATION TREES

Let $\mathbb{T} = \{\mathcal{T}_1, \cdots, \mathcal{T}_n\}$ be a set of computation trees. The *frequency* of $\mathbb{T}$ in the train set $\mathcal{D} = \{\mathcal{G}_1, \cdots, \mathcal{G}_m\}$ is defined as the proportion of graphs that contain all of the computation trees in $\mathbb{T}$. Formally,

$$freq(\mathbb{T}) = \frac{\left|\{\mathcal{G} \in \mathcal{D} \mid \forall \mathcal{T} \in \mathbb{T}, \exists \mathcal{T}_v^L \in \mathbb{T}_\mathcal{G}, \mathcal{T} \cong \mathcal{T}_v^L\}\right|}{|\mathcal{D}|} \quad (6)$$

Here, $\mathbb{T}_\mathcal{G}$ denotes the set of computation trees in graph $\mathcal{G}$.

**Problem 2** (Mining Frequent Co-occurring Trees). *Given a set of $|\mathcal{D}|$ computation tree multi-sets[4] $\mathfrak{T} = \{\mathbb{T}_1, \cdots, \mathbb{T}_m\}$ corresponding to each graph in the train set $\mathcal{D} = \{\mathcal{G}_1, \cdots, \mathcal{G}_m\}$, and a threshold $\theta$, mine all co-occurring trees with frequency of at least $\theta$. Formally, we seek to identify the following distilled answer set.*

$$\mathcal{S} = \{\mathcal{X} \subset \mathcal{I} \mid freq(\mathcal{X}) \geq \theta\} \text{ where } \mathcal{I} = \bigcup_{\forall \mathbb{T}_i \in \mathfrak{T}} \{\mathbb{T}_i\} \quad (7)$$

*$\mathcal{I}$ denotes the universe of all unique computation trees, i.e., $\forall \mathcal{T}_i, \mathcal{T}_j \in \mathcal{I}, \mathcal{T}_i \not\cong \mathcal{T}_j$.*

We map Problem 2 to the problem of mining frequent itemsets from transaction databases (Han et al., 2004), which we solve using the FPGROWTH algorithm (Han et al., 2004).

## 3.4 MODELING AND INFERENCE

Algorithm. 1 in the appendix outlines the pseudocode of our data distillation and Algorithm. 2 outlines the modeling algorithm. We decompose each graph into their computation trees. We mine the frequently co-occurring trees from each class separately. Instead of training on a batch of graphs, we sample a batch of frequent tree sets. Each of these frequent tree sets serves as a surrogate for an entire graph, allowing us to approximate the graph embedding. To achieve this approximation, we utilize the COMBINE function (Eq. 5) on the embeddings of the root node within each tree present in the selected set. The probability of selecting a particular tree set for sampling is directly proportional to its frequency of occurrence.

## 3.5 PROPERTIES AND PARAMETERS

**Parameters:** As opposed to existing graph distillation algorithms (Jin et al., 2022; 2021; Xu et al., 2023), which are dependent on the specific choice of GNN architecture and all hyper-parameters that the GNN relies on, MIRAGE intakes only two parameters: the number of GNN layers $L$ and the frequency threshold $\theta$. $\theta$, which lies in $[0, 1]$, is a GNN independent parameter. The size of the distilled dataset increases monotonically with decrease in $\theta$. Hence, $\theta$ may be selected based on the desired distillation size. $L$ is the only model-specific information we require. We note that the number of

---

[4]non-isomorphic graphs may decompose to the same set of computation trees

layers used while training needs to be $\leq L$, and need not exactly $L$, since $\mathcal{T}_v^L \supseteq \mathcal{T}_v^{L-1}$. Hence, $L$ should be set based on the expected upper limit that may be used. GNNs are typically run with $L \leq 3$ due to the well-known issue of over-smoothing and over-squashing (Topping et al., 2022).

**Algorithm Characterization:** MIRAGE has several salient characteristics when compared to existing baselines, all arising due to being unsupervised to original training gradients-the predominant approach in graph distillation.

- **Robustness:** The distillation process is independent of training hyper-parameters (except the mild assumption on maximum number of GNN layers) and choice of GNN architecture. Hence, it does not need to be regenerated for changes to any of the above factors.
- **Storage Overhead:** MIRAGE has a smaller storage footprint since a single distilled dataset suffices for all combinations of architecture and hyper-parameters.
- **CPU-bound executions and efficiency:** The distillation pipeline is a function of the training dataset only. Hence, it is computationally efficient requiring only CPU-bound operations.

**Complexity Analysis:** A detailed complexity analysis of MIRAGE is provided in Appendix A. We also discuss strategies to speed-up tree frequency counting through the usage of *canonical labels*. In summary, the entire process of decomposing the full graph database into computation tree sets incurs $\mathcal{O}(z \times \delta^L)$ cost, where $z = \sum_{\forall \mathcal{G} \in \mathcal{D}} |\mathcal{V}|$ and $\delta$ is the average degree of nodes. Counting frequency of all trees consume $\mathcal{O}\left(z \times L\delta^L \log(\delta)\right)$ time. FPGROWTH consumes $\mathcal{O}(2^{|\mathcal{I}|})$ in the worst case, but it has been shown in the literature that empirical efficiency is dramatically faster due to sparsity in frequent patterns (Han et al., 2004).

## 4 EXPERIMENTS

In this section, we benchmark MIRAGE and establish:

- **Accuracy:** MIRAGE is the most robust distillation algorithm and consistently ranks among the top-2 performers across all dataset-GNN combinations.
- **Compression:** MIRAGE achieves the highest compression on average, which is $\approx 4$ and $\approx 5$ times smaller that the state of the art algorithms of DOSCOND and KIDD respectively.
- **Efficiency:** MIRAGE is $\approx 150$ and $\approx 500$ times faster than DOSCOND and KIDD on average.

All experiments have been executed 5 times. We report the mean and standard deviations. The codebase of MIRAGE is shared at https://github.com/idea-iitd/Mirage. For details on the hardware and software platform used, please refer to Appendix B.1.

### 4.1 DATASETS

To evaluate MIRAGE, we use datasets from *Open Graph Benchmark* (OGB) (Hu et al., 2020) and *TU Datasets* (DD, IMDB-B and NCI1) (Morris et al., 2020) spanning a variety of domains. The chosen datasets represent sufficient diversity in graph sizes ($\approx$

Table 1: Dataset statistics

| Dataset | #Classes | #Graphs | Avg. Nodes | Avg. Edges | Domain |
|---|---|---|---|---|---|
| ogbg-molbace | 2 | 1513 | 34.1 | 36.9 | Molecules |
| NCI1 | 2 | 4110 | 29.9 | 32.3 | Molecules |
| ogbg-molbbbp | 2 | 2039 | 24.1 | 26.0 | Molecules |
| ogbg-molhiv | 2 | 41,127 | 25.5 | 54.9 | Molecules |
| DD | 2 | 1178 | 284.3 | 715.7 | Proteins |
| IMDB-B | 2 | 1000 | 19.39 | 193.25 | Movie |
| IMDB-M | 3 | 1500 | 13 | 65.1 | Movie |

24 nodes to $\approx 284$ nodes) and density ($\approx 1$ to $\approx 10$).

### 4.2 EXPERIMENTAL SETUP

**Baselines.** Among neural baselines, we consider the state of the art graph distillation algorithms for graph classification, which are **(1)** DOSCOND (Jin et al., 2022) and **(2)** KIDD (Xu et al., 2023). We do not consider GCOND (Jin et al., 2021) since DOSCOND have been shown to consistently outperform GCOND. KIDD supports graph distillation only GIN. We also include **(3)** HERDING (Welling, 2009) maps graphs into embeddings using the target GNN architecture. Subsequently, it selects the graphs that are closest to the cluster centers in the distilled set. Finally, we consider the **(4)** RANDOM baseline, wherein we randomly select graphs over iterations from each class in the dataset till the combined size exceeds the size of the distilled dataset produced by MIRAGE.

**Evaluation Protocol.** We benchmark MIRAGE and considered baselines across three different GNN architectures, namely GCN (Kipf & Welling, 2016), GAT (Veličković et al., 2018) and GIN (Xu et al., 2019). It is worth noting that this is the first graph distillation study to span three GNN architectures when compared DOSCOND or KIDD, that evaluate only on a specific GNN of choice. KIDD only

Table 2: AUC-ROC of benchmarked algorithms across datasets and GNN architectures. The best and the second best AUC-ROC in each dataset is highlighted in dark and light green colors respectively. We do not report the results of GAT in IMDB-B and IMDB-M since GAT achieves an AUC-ROC of $\approx 0.5$ across the full datasets and their distilled versions for all baselines. These datasets do not contain any node features and GAT struggles to learn attention in this scenario.

| Dataset | Model | RANDOM (mean) | RANDOM (sum) | HERDING | KIDD | DOSCOND | MIRAGE | Full Dataset |
|---|---|---|---|---|---|---|---|---|
| ogbg-molbace | GAT | 65.43 ± 3.57 | 73.75 ± 2.30 | 58.39 ± 7.04 | 66.16 ± 4.62 | 68.30 ± 1.01 | 70.77 ± 1.67 | 77.20 ± 2.20 |
| | GCN | 62.96 ± 3.25 | 76.03 ± 0.60 | 52.46 ± 6.47 | 63.92 ± 13.1 | 67.34 ± 1.84 | 77.03 ± 1.24 | 77.31 ± 1.60 |
| | GIN | 57.18 ± 10.4 | 74.95 ± 2.28 | 65.24 ± 6.17 | 77.09 ± 0.57 | 63.41 ± 0.66 | 76.18 ± 0.61 | 78.53 ± 3.70 |
| NCI1 | GAT | 50.46 ± 2.65 | 64.01 ± 6.87 | 66.77 ± 1.11 | 60.62 ± 1.47 | 58.10 ± 1.52 | 68.10 ± 0.20 | 83.50 ± 0.71 |
| | GCN | 51.36 ± 0.36 | 60.72 ± 8.06 | 66.86 ± 0.73 | 64.85 ± 2.32 | 57.90 ± 0.75 | 68.20 ± 0.04 | 87.03 ± 0.57 |
| | GIN | 51.60 ± 5.85 | 61.15 ± 7.30 | 67.12 ± 1.90 | 60.83 ± 2.26 | 59.80 ± 2.30 | 67.91 ± 0.31 | 85.60 ± 2.19 |
| ogbg-molbbbp | GAT | 57.16 ± 2.20 | 60.40 ± 1.84 | 59.15 ± 4.13 | 62.88 ± 3.31 | 61.12 ± 2.51 | 63.05 ± 1.10 | 64.70 ± 2.10 |
| | GCN | 60.18 ± 2.66 | 58.76 ± 3.51 | 55.93 ± 1.09 | 58.77 ± 1.83 | 59.19 ± 0.95 | 61.30 ± 0.52 | 64.43 ± 2.21 |
| | GIN | 60.06 ± 3.85 | 60.21 ± 3.14 | 54.88 ± 2.84 | 64.21 ± 0.99 | 61.10 ± 2.10 | 61.21 ± 0.77 | 64.95 ± 2.24 |
| ogbg-molhiv | GAT | 53.35 ± 4.78 | 64.61 ± 8.43 | 61.82 ± 1.75 | 69.79 ± 0.64 | 72.33 ± 0.85 | 73.10 ± 0.96 | 73.71 ± 1.36 |
| | GCN | 48.21 ± 5.95 | 67.20 ± 6.16 | 59.36 ± 2.79 | 69.56 ± 2.74 | 73.16 ± 0.69 | 69.59 ± 3.29 | 75.93 ± 1.29 |
| | GIN | 53.07 ± 7.07 | 69.94 ± 1.42 | 69.66 ± 2.64 | 63.02 ± 4.48 | 72.72 ± 0.80 | 71.58 ± 1.42 | 78.66 ± 1.31 |
| DD | GAT | 50.87 ± 1.10 | 67.31 ± 12.0 | 71.20 ± 2.14 | 73.14 ± 4.32 | 63.45 ± 2.47 | 76.08 ± 0.63 | 76.36 ± 0.09 |
| | GCN | 53.58 ± 2.38 | 58.02 ± 8.57 | 65.26 ± 5.63 | 71.04 ± 6.04 | 68.39 ± 9.64 | 74.84 ± 2.15 | 75.37 ± 1.23 |
| | GIN | 57.34 ± 1.60 | 67.50 ± 9.66 | 73.23 ± 3.62 | 64.55 ± 3.50 | 60.23 ± 1.76 | 74.45 ± 0.67 | 74.74 ± 0.58 |
| IMDB-B | GCN | 52.06 ± 4.90 | 50.38 ± 0.31 | 60.69 ± 3.43 | 58.29 ± 0.61 | 55.56 ± 2.83 | 59.17 ± 0.07 | 60.84 ± 2.50 |
| | GIN | 51.31 ± 4.10 | 51.12 ± 2.76 | 60.48 ± 3.28 | 57.45 ± 0.16 | 60.02 ± 2.49 | 62.18 ± 0.17 | 66.73 ± 1.53 |
| IMDB-M | GCN | 55.10 ± 3.80 | 52.90 ± 2.52 | 61.00 ± 2.40 | 57.10 ± 1.11 | 55.90 ± 1.06 | 63.20 ± 1.12 | 64.10 ± 1.10 |
| | GIN | 60.10 ± 2.67 | 56.30 ± 5.50 | 58.47 ± 4.12 | 54.18 ± 0.90 | 58.30 ± 1.70 | 61.80 ± 1.51 | 64.80 ± 1.10 |

supports GIN. Hence, for other GNN architectures, we use the distilled dataset for GIN, but train using the target GNN.

**Parameter settings.** Hyper-parameters used to train MIRAGE, the baselines, and the GNN models are discussed in Appendix B.2.

## 4.3 PERFORMANCE IN GRAPH DISTILLATION

**Prediction Accuracy.** In Table 2, we report the mean and standard deviation of the testset AUC-ROC of all baselines on the distilled dataset as well as the AUC-ROC when trained on the full dataset. Several important insights emerge from Table 2.

Firstly, it is noteworthy that MIRAGE consistently ranks as either the top performer or the second-best across all combinations of datasets and architectures. Particularly striking is the fact that MIRAGE achieves the best performance in 8 out of the 17 dataset-architecture combinations, which stands as the highest number of top rankings among all considered baselines. This demonstrates that being unsupervised to original training gradients does not hurt MIRAGE's prediction accuracy.

Secondly, we observe instances, such as in DD, where the distilled dataset outperforms the full dataset, an outcome that might initially seem counter-intuitive. This phenomenon has been reported in the literature before (Xu et al., 2023). While pinpointing the exact cause behind this behavior is challenging, we hypothesize that the distillation process may tend to remove outliers from the training set, subsequently leading to improved accuracy. Additionally, given that distillation prioritizes the selection of graph components that are more informative to the task, it is likely to retain the most critical patterns, resulting in enhanced model generalizability.

Finally, we note that the performance of RANDOM (sum), which involves random graph selection and the COMBINE function (Eq. 5) being SUMPOOL, is surprisingly strong, and at times surpassing the performance of all baselines. Interestingly, in the literature, DOSCOND and KIDD have reported results only with RANDOM (mean), which is substantially weaker. We investigated this phenomenon and noticed that in datasets where RANDOM (sum) performs well, the label distribution of nodes and the number of nodes across the classes are noticeably different. SUMPOOL is better at preserving these magnitude differences in node and label counts compared to MEANPOOL, which averages them out.

**Compression.** We next investigate the size of the distilled dataset. MIRAGE is independent of the underlying GNN architecture, ensuring that its size remains consistent regardless of the specific architecture employed. On the other hand, KIDD, as previously indicated in § 4.2, conducts distillation with the assumption that GIN serves as the underlying GNN architecture. In the case of DOSCOND and HERDING, these methods support various GNN architectures; however, the size of the distilled datasets is architecture-specific for each. It is important to note that we exclude RANDOM from this analysis as, per our discussion in § 4.2, we select graphs until the dataset's size exceeds that of MIRAGE. Consequently, by design, its size closely aligns with that of MIRAGE.

Table 3: Size of distilled dataset, in terms of *bytes*, produced by benchmarked algorithms across datasets and GNN architectures. The best compression is highlighted in dark green color. The results for IMDB-B and IMDB-M for GAT represented as - are skipped since GAT achieves $\approx 0.5$ AUC-ROC on IMDB-B and IMDB-M.

| Method → Dataset ↓ | HERDING | | | KiDD | DOSCOND | | | MIRAGE | Full Dataset |
|---|---|---|---|---|---|---|---|---|---|
| | GAT | GCN | GIN | | GAT | GCN | GIN | | |
| ogbg-molbace | 25,771 | 26,007 | 26,129 | 2,592 | 23,176 | 23,176 | 23,176 | 1,612 | 1,610,356 |
| NCI1 | 5,662 | 5,680 | 5,683 | 26,822 | 70,168 | 70,760 | 73,128 | 318 | 1,046,828 |
| ogbg-molbbbp | 10,497 | 10,514 | 10,466 | 3,618 | 13,632 | 14,280 | 20,832 | 6,108 | 740,236 |
| ogbg-molhiv | 21,096 | 21,096 | 21,140 | 7,672 | 4,808 | 5,280 | 4,400 | 3,288 | 41,478,694 |
| DD | 89,882 | 89,869 | 90,086 | 408,980 | 210,168 | 210,184 | 209,816 | 448 | 7,414,218 |
| IMDB-B | - | 1,238 | 1,252 | 980 | - | 1184 | 2484 | 280 | 635,856 |
| IMDB-M | - | 1,156 | 1,256 | 936 | - | 720 | 824 | 228 | 645,160 |

In Table 3, we present the compression results. MIRAGE stands out by achieving the highest compression in 5 out of 6 datasets. In the single dataset where it does not hold the smallest size, MIRAGE still ranks as the second smallest, showcasing its consistent compression performance. On average, MIRAGE achieves a compression rate that is $\approx 4$ times higher compared to DOSCOND and 5 times greater than KiDD. This notable advantage of MIRAGE over the baseline methods underscores the effectiveness of exploiting data distribution over replicating gradients, at least within the context of graph databases where recurring patterns are prevalent.

**Distillation Time.** We now focus on the efficiency of the distillation process. Fig. 4a presents this information. We observe that MIRAGE is more than $\approx 500$ times faster on average than KiDD and $\approx 150$ times faster than DOSCOND. This impressive computational-efficiency is achieved despite MIRAGE utilizing only a CPU for its computations, whereas DOSCOND and KiDD are reliant on GPUs. This trend is a direct consequence of MIRAGE not being dependent on training on the full data. KiDD is slower than DOSCOND since, while both seek to replicate the gradient trajectory of model weights, KiDD solves this optimization problem exactly, whereas DOSCOND is an approximation. When compared to the training time on full dataset (See Table I, MIRAGE is more than 30 times faster on average). Overall, MIRAGE is not only faster, but also presents a more environment-friendly and energy-efficient approach to graph distillation.

## 4.4 SUFFICIENCY OF FREQUENT TREE PATTERNS

In order to establish the sufficiency of frequent tree patterns in capturing the dataset characteristic, we conduct the following experiment. We train the model on the full dataset and store its weights at each epoch. Then, we freeze the model at the weights after each epoch's training and pass both the distilled dataset consisting of just the frequent tree patterns and the full dataset. We then compute the differences between the losses as shown in Fig. 5a. We do this for all the models for datasets ogbg-molbace, ogbg-molbbbp, and ogbg-molhiv (full results in Figure I in appendix). The rationale behind this is that the weights of the full model recognise the patterns that are important towards minimizing the loss. Now, if the same weights continue to be effective on the distilled train set, it indicates that the distilled dataset has retained the important information. In the figure, we can see that the difference quickly approaches 0 for all the models for all the datasets, and only starts at a high value at the random initialization where the weights are not yet trained to recognize the important patterns. Furthermore, gradient descent will run more iterations on trees that it sees more often and hence infrequent trees have limited impact on the gradients. Further, in Fig. 5b, we plot the train loss on full and distilled dataset with their own parameters learned through independent training. As visible, the losses are similar, further substantiating the rich information content in

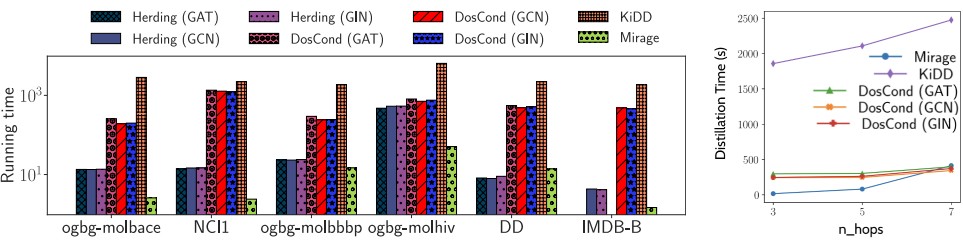

(a) Distillation times for the different methods. Full numbers and standard deviations are in Table E in Appendix.

(b) Distillation time vs number of hops

Figure 4: (a) Distillation times for the different methods. (b) Distillation time as a function of number of hops on ogbg-molbbbp dataset.

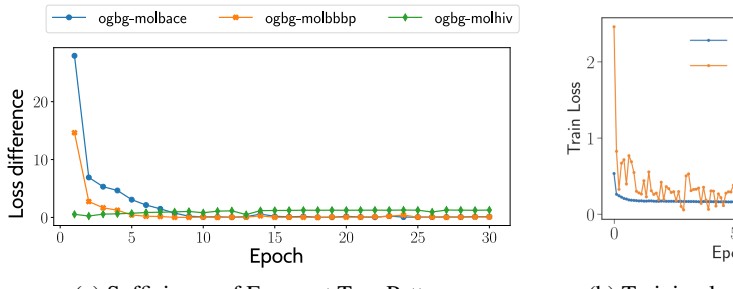

(a) Sufficiency of Frequent Tree Patterns     (b) Training loss vs epochs

Figure 5: (a) For this experiment the model weights are extracted after each epoch. Then, the model weights are loaded from the epoch weights and kept fixed for the following procedure. The dataset condensed using MIRAGE and the full dataset are then passed through the model. The difference between the losses is plotted. The difference between the losses approaches 0. Note that the model was trained on the full dataset. (b) Training loss vs epochs on ogbg-molhiv(GCN). Results on more datasets can be found in Appendix H.

the distilled dataset. These results empirically establish the sufficiency of frequent tree patterns in capturing the majority of the dataset characteristics.

## 4.5 IMPACT OF PARAMETERS

**Impact of Frequency Threshold.** In Appendix F we study the impact of frequency threshold on distillation efficiency.

**Impact of Number of Hops:** In Appendix 4b we analyze the efficiency of distillation as the number of hops increase. We observe running time of MIRAGE is lower or similar to other distillation methods as number of hops increase. For more details see Appendix F.

We refer the reader to Appendix F and H for more experiments on parameter variations and their impact on AUC and efficiency.

## 5 CONCLUSIONS, LIMITATIONS AND FUTURE WORKS

Training Graph Neural Networks (GNNs) on large-scale graph datasets can be computationally intensive and resource-demanding. To address this challenge, one potential solution is to distill the extensive graph dataset into a more compact synthetic dataset while maintaining competitive predictive accuracy. While the concept of graph distillation has gained attention in recent years, existing methods typically rely on model-related information, such as gradients or embeddings. In this research endeavor, we introduce a novel framework named MIRAGE, which employs a frequent pattern mining-based approach. MIRAGE leverages the inherent design of message-passing frameworks, which decompose graphs into computation trees. It capitalizes on the observation that the distribution of these computation trees often exhibits a highly skewed nature. This unique feature enables us to compress the computational data itself without requiring access to specific model details or hyper-parameters, aside from a reasonable assumption regarding the maximum number of GNN layers. Our extensive experimentation across six real-world datasets, in comparison to state-of-the-art algorithms, demonstrates MIRAGE's superiority across three critical metrics: predictive accuracy, a distillation efficiency that is 150 times higher, and data compression rates that are 4 times higher. Moreover, it's noteworthy that MIRAGE solely relies on CPU-bound operations, offering a more environmentally sustainable alternative to existing algorithms.

**Limitations and Future Works:** MIRAGE, as well as, existing graph distillation algorithms currently lack the ability to generalize effectively to unseen tasks. Moreover, their applicability to other types of graphs, such as temporal networks, remains unexplored. Additionally, there is a need to assess how these existing algorithms perform on contemporary architectures like graph transformers (e.g., (Ying et al., 2021; Rampášek et al., 2022)) or equivariant GNNs (e.g., (Satorras et al., 2022)). Our future work will be dedicated to exploring these avenues of research. Finally, MIRAGE relies on the assumption that the distribution of computation trees is skewed. Although we provide compelling evidence of its prevalence across a diverse range of datasets, this assumption may not hold universally, especially in the case of heterophilous datasets. The development of a model-agnostic distillation algorithm remains an open challenge in such scenarios.

## 6 ACKNOWLEDGEMENT

We acknowledge the Yardi School of AI, IIT Delhi for supporting this research. Mridul Gupta acknowledges Google for supporting his travel. Sahil Manchanda acknowledges GP Goyal Alumni Grant of IIT Delhi for supporting his travel.

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

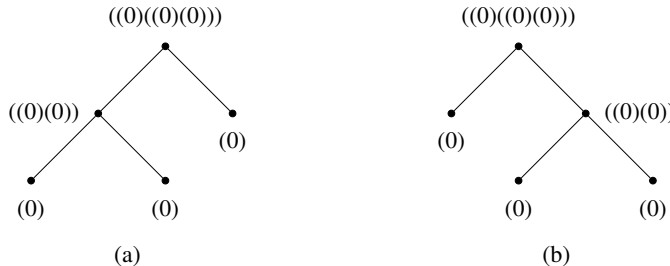

Figure F: (a) and (b) show Knuth Tuples based canonical labels for two isomorphic trees. The process starts at the leaves and goes up to the root. Whenever a node encapsulates its children's labels, it sorts them in increasing order of length. This can be adapted for the cases when nodes have labels and when the edges have labels by making appropriate changes to the tuples.

## APPENDIX

### A    COMPLEXITY ANALYSIS

**Computation tree decomposition:** Each graph $\mathcal{G} = (\mathcal{V}, \mathcal{E}, \boldsymbol{X})$, decomposes into $|\mathcal{V}|$ computation trees. Assuming an average node degree of $\delta$, enumerating a computation tree consumes $\mathcal{O}(\delta^L)$ time. Hence, the entire process of decomposing the full graph database into computation tree sets incurs $\mathcal{O}(z \times \delta^L)$ computation cost, where $z = \sum_{\forall \mathcal{G} \in \mathcal{D}} |\mathcal{V}|$.

**Frequency counting:** Computing the frequency of a computation tree requires us to perform tree isomorphism test. Although no polynomial time algorithm exists for graph isomorphism, in rooted trees, it can be performed in linear time to the number of nodes in the tree (Valiente, 2002), which in our context is $\mathcal{O}(\delta^L)$. Thus, frequency counting of all trees requires $\mathcal{O}(\delta^L \times z^2)$ time. In MIRAGE, we optimize frequency counting further using *canonical labeling* (Campbell & Radford, 1991).

**Definition 4** (Canonical label). *A canonical label of a graph $\mathcal{G}$ involves defining a unique representation or labeling of a graph in a way that is invariant under isomorphism. Specifically, if $\mathcal{L}$ is the function that maps a graph to its canonical label, then*

$$\mathcal{G}_1 \cong \mathcal{G}_2 \iff \mathcal{L}(\mathcal{G}_1) = \mathcal{L}(\mathcal{G}_2)$$

There are several algorithms available described in (Campbell & Radford, 1991) and (Chi et al., 2005) that map rooted-trees to canonical labels. We use (Campbell & Radford, 1991) in our implementation, which is explained in Fig. F.

Canonical label construction for a rooted tree consumes $\mathcal{O}(m \log(m))$ time if the tree contains $m$ nodes. In our case, $m = \mathcal{O}(\delta^L)$ as discussed earlier. Thus, the complexity is $\mathcal{O}(\delta^L \log(\delta^L)) = \mathcal{O}(L\delta^L \log(\delta))$ time. Once trees have been constructed, frequency counting involves hashing each of the canonical labels, which takes linear time to the number of graphs. Hence, the complexity reduces to $\mathcal{O}(z \times L\delta^L \log(\delta))$ when compared to the all pairs tree isomorphism approach of $\mathcal{O}(\delta^L \times z^2)$ ($L \log(\delta) \ll z$).

**Frequent itemset mining:** Finally, in the frequent itemset mining step, the complexity in the worst case is $\mathcal{O}(2^{|\mathcal{I}|})$. In reality, however, the running times are dramatically smaller due to majority of items (trees in our context) being infrequent (and hence itemsets as well) (Han et al., 2004).

### B    EMPIRICAL SETUP

#### B.1    HARDWARE AND SOFTWARE PLATFORM

All experiments are performed on an Intel Xeon Gold 6248 processor with 96 cores and 1 NVIDIA A100 GPU with 40GB memory, and 377 GB RAM with Ubuntu 18.04. In all experiments, we have trained using the Adam optimizer with a learning rate of 0.0001 and choose the model based on the best validation loss.

---

**Algorithm 1** MIRAGE: Proposed graph distillation algorithm

---

    **Input** Train set $\mathcal{D}$, number of layers $L$ in GNN, frequency threshold $\theta$.
    **Output** Distilled dataset $\mathcal{S}$ and parameters $\Theta$ of the GNN when trained on $\mathcal{S}$
 1: $\mathcal{S} \leftarrow \emptyset$
 2: **for each** Class $c$ in dataset **do**
 3:     $\mathfrak{T}_c \leftarrow \emptyset$
 4:     **for each** $\mathcal{G} = (\mathcal{V}, \mathcal{E}, \boldsymbol{X}) \in \mathcal{D}$ such that $\mathcal{Y}_{\mathcal{G}} = c$ **do**
 5:         $\mathbb{T} \leftarrow \emptyset$
 6:         **for each** $v \in \mathcal{V}$ **do**
 7:             $\mathbb{T} \leftarrow \mathbb{T} \cup \left\{ \mathcal{T}_v^L = \text{compute-tree}(\mathcal{G}, v, L) \mid \forall v \in \mathcal{V} \right\}$
 8:         $\mathfrak{T}_c \leftarrow \mathfrak{T}_c \cup \mathbb{T}$
 9:     $\mathcal{S} \leftarrow \mathcal{S} \cup \text{FPGROWTH}(\mathfrak{T}_c, \theta)$
10: **Return** $\mathcal{S}$

---

**Algorithm 2** Training a GNN using data distilled using MIRAGE

---

    **Input** Distilled dataset $\mathcal{S}$
    **Output** Parameters $\Theta$ of the GNN when trained on $\mathcal{S}$
 1: Randomly initialize $\Theta$
 2: **while** Model loss has not converged **do**
 3:     $\mathfrak{B} \leftarrow$ a batch of tree sets sampled in proportion to their frequencies from $\mathcal{S}$
 4:     **for each** $\mathbb{T} \in \mathfrak{B}$ **do**
 5:         $\mathbf{h}_{\mathbb{T}} \leftarrow \text{COMBINE}(\mathbf{h}_v^L, \forall \mathcal{T}_v^L \in \mathbb{T})$
 6:     Update $\Theta$ using backpropagation based on loss over $\{\mathbf{h}_{\mathbb{T}} \mid \forall \mathbb{T} \in \mathfrak{B}\}$
 7: **Return** $\Theta$

---

### B.2 PARAMETERS

Table Da presents the parameters used to train MIRAGE. Note that the same distillation parameters are used for all benchmarked GNN architectures and hence showcasing its robustness to different flavors of modeling pipelines.

For neural baselines KIDD and DOSCOND, we use the same parameters recommended in their respective papers on datasets that are also used in their studies. Otherwise, the optimal parameters are chosen using grid search.

For the model hyper-parameters, we perform grid search to optimize performance on the whole dataset. The same parameters are used to train and infer on the distilled dataset. The hyper-parameters used are shown in Table Db.

**Train-validation-test Splits.** The OGB datasets come with the train-validation-test splits, which are also used in DOSCOND and KIDD. For TU Datasets, we randomly split the graphs into $80\%/10\%/10\%$ for training-validation-test. We stop the training of a model if it does not improve the validation loss for more than 15 epochs.

Table D: Parameters used for MIRAGE.

(a) Distillation parameters. $\theta_0$ and $\theta_1$ represent the frequency thresholds in class 0 and 1 respectively.

| Dataset | $\theta_0$ | $\theta_1$ | #hops ($L$) |
|---|---|---|---|
| NCI1 | 27% | 35% | 2 |
| ogbg-molbbbp | 5% | 7% | 2 |
| ogbg-molbace | 13% | 10% | 3 |
| ogbg-molhiv | 5% | 8% | 3 |
| DD | 2% | 2% | 1 |
| IMDB-B | 20% | 20% | 1 |

(b) Model parameters

| Model | Layers | Hidden Dimension | Dropout | Reduce Type |
|---|---|---|---|---|
| GCN | $\{2,3\}$ | $\{64, 128\}$ | $[0, 0.6]$ | {sum,mean} |
| GAT | $\{2,3\}$ | $\{64, 128\}$ | $[0, 0.6]$ | {sum,mean} |
| GIN | $\{2,3\}$ | $\{64, 128\}$ | $[0, 0.6]$ | {sum,mean} |

Table E: Graph distillation time (in seconds) consumed by various algorithms. In the last column, we also present the total training time in the full dataset to showcase the efficiency gain of distillation.

| Method → | HERDING | | | DOSCOND | | | KIDD | MIRAGE |
|---|---|---|---|---|---|---|---|---|
| Dataset ↓ | GAT | GCN | GIN | GAT | GCN | GIN | | |
| ogbg-molbace | $13.47 \pm 0.52$ | $13.41 \pm 0.57$ | $13.62 \pm 0.72$ | $255.62 \pm 7.52$ | $191.22 \pm 5.62$ | $198.89 \pm 6.21$ | 2839.40 | $2.57 \pm 0.12$ |
| NCI1 | $13.86 \pm 0.29$ | $14.64 \pm 0.48$ | $14.70 \pm 0.43$ | $1348.21 \pm 11.2$ | $1275.82 \pm 13.2$ | $1237.98 \pm 82.5$ | 2200.04 | $2.39 \pm 0.16$ |
| ogbg-molbbbp | $23.35 \pm 1.59$ | $23.18 \pm 1.33$ | $23.86 \pm 1.32$ | $295.02 \pm 3.34$ | $240.91 \pm 7.58$ | $244.44 \pm 5.33$ | 1855.81 | $14.78 \pm 0.09$ |
| ogbg-molhiv | $473.44 \pm 23.8$ | $530.89 \pm 27.8$ | $535.08 \pm 32.74$ | $808.11 \pm 42.3$ | $708.33 \pm 7.54$ | $755.48 \pm 54.9$ | 6421.98 | $50.86 \pm 0.32$ |
| DD | $8.18 \pm 0.51$ | $7.95 \pm 0.57$ | $9.02 \pm 0.56$ | $551.39 \pm 8.36$ | $485.81 \pm 3.79$ | $511.14 \pm 4.13$ | 2201.09 | $13.93 \pm 0.16$ |
| IMDB-B | - | $4.31 \pm 0.52$ | $4.14 \pm 0.63$ | - | $482.01 \pm 4.21$ | $455.02 \pm 3.98$ | 1841.80 | $1.47 \pm 0.01$ |

Table F: **Cross-arch performance:** The performance of DOSCOND when graph distillation is performed using gradients of a particular GNN, while the model is trained on another GNN.

| train+test→ condensed using↓ | GAT | GCN | GIN |
|---|---|---|---|
| GAT | $\mathbf{61.12 \pm 2.51}$ | $\mathbf{59.86 \pm 1.50}$ | $59.36 \pm 1.26$ |
| GCN | $59.33 \pm 3.37$ | $59.19 \pm 0.95$ | $57.02 \pm 3.09$ |
| GIN | $58.20 \pm 1.91$ | $56.42 \pm 1.68$ | $\mathbf{61.10 \pm 2.10}$ |

(a) ogbg-molbbbp

| train+test→ condensed using↓ | GAT | GCN | GIN |
|---|---|---|---|
| GAT | $\mathbf{68.30 \pm 1.01}$ | $67.01 \pm 4.21$ | $62.90 \pm 4.84$ |
| GCN | $63.70 \pm 2.98$ | $\mathbf{67.34 \pm 1.84}$ | $58.91 \pm 4.85$ |
| GIN | $65.7 \pm 3.81$ | $66.47 \pm 3.83$ | $\mathbf{63.41 \pm 0.66}$ |

(b) ogbg-molbace

## C   DISTILLATION GENERALIZATION OF DOSCOND

While message-passing GNNs come in various architectural forms, one may argue that the embeddings generated, when the data and the loss are same, are correlated. Hence, even in the case of GNN-dependent distillation algorithms, such as DOSCOND, it stands to reason that the same distillation data could generalize well to other GNNs. In Table F, we investigate this hypotheses. Across the six evaluated combinations, except for the case of GCN in ogbg-molbbbp, we consistently observe that the highest performance is achieved when the distillation GNN matches the training GNN. This behavior is unsurprising since although GNNs share the initial task of breaking down input graphs into individual components of message-passing trees, subsequent computations diverge. For instance, GIN employs SUMPOOL, which is density-dependent and retains magnitude information. Conversely, GCN, owing to their normalization based on node degrees, does not preserve magnitude information as effectively. GAT, on the other hand, utilizes attention mechanisms, resulting in varying message weights learned as a function of the loss. In summary, Table F provides additional evidence supporting the necessity for GNN-independent distillation algorithms.

Table G: Below we present the AUCROC numbers of DOSCOND on randomly initialized GCN models, warm start (100 epochs) and converged DOSCOND (typically around 1000 epochs).

| Dataset | Random | Warm Started | Convergence |
|---|---|---|---|
| ogbg-molbace | $55.04 \pm 9.07$ | $59.95 \pm 1.61$ | $67.34 \pm 1.84$ |
| NCI1 | $51.22 \pm 2.00$ | $48.18 \pm 2.78$ | $57.90 \pm 0.75$ |
| ogbg-molbbbp | $52.64 \pm 1.98$ | $50.72 \pm 3.48$ | $59.19 \pm 0.95$ |
| ogbg-molhiv | $48.21 \pm 5.95$ | $34.99 \pm 7.25$ | $73.16 \pm 0.69$ |
| DD | $52.39 \pm 7.19$ | $61.58 \pm 2.11$ | $68.39 \pm 9.64$ |

Table H: Node classification results

| Dataset Type | Size (bytes) | AUC-ROC (%) |
|---|---|---|
| Full | 1610356 | $91.15 \pm 0.09$ |
| Distilled using MIRAGE | 1816 | $\mathbf{88.92 \pm 0.77}$ |
| Distilled using DOSCOND | 7760 | $81.29 \pm 3.81$ |

## D    DOSCOND DISTILLATION: RANDOM, WARM-STARTED AND FULLY OPTIMIZED

In this section we investigate how the quality of condensed dataset synthesized by DOSCOND changes during its course of optimization. Towards this, we obtain the condensed dataset at random initialization, after optimizing for small number of epochs and after training of DOSCOND until convergence. In Table G, we present the results. As visible, there is a noticeable gap in the AUCROC numbers indicating full training is necessary.

## E    NODE CLASSIFICATION

The primary focus of MIRAGE is on graph classification. However, it can be easily extended to node classification. Specifically, we omit the graph level embedding construction (Eq. 5) and training is performed on node level embeddings (Eq. 4). We use ogbg-molbace to analyze performance in node classification. Here, each node is labeled as aromatic or non-aromatic depending on whether it is part of an aromatic ring substructure. The results are tabulated in Table H. Consistent with previous results, MIRAGE outperforms DOSCOND in both AUC-ROC and compression (smaller size of the distilled dataset). DOSCOND produces a dataset that is $\approx 4$ times the size of that produced by MIRAGE, yet performs more than $7\%$ lower in AUC-ROC. This coupled with model-agnostic-ness further solidifies the superiority of MIRAGE.

## F    COMPUTATIONAL COST OF DISTILLATION

In clarifying the computational overhead inherent in the dataset distillation procedure, we conduct a series of experiments. Initially, we manipulate the number of hops, recording the corresponding distillation time (Figure H). Simultaneously, we provide training time metric for the full dataset setting (Table I), facilitating a comparative analysis. Our findings reveal that even under high hop counts from the GNN perspective, the distillation process is more time-efficient than complete dataset training. Moreover, the distilled dataset's performance converges closely with that of the full dataset, as evident in Table 2.



|              (a) DD              |              (b) IMDB-B              |              (c) ogbg-molhiv              |

Figure G: Impact of frequency threshold (both positive and negative classes) on the distillation time. Here, the thresholds on the positive and negative classes are varied in the $y$ and $x$ axis respectively, and the time is presented as a contour.

Subsequently, we subject the system to variations in threshold parameters, graphically showing the resulting time in Figure G. Notably, the distillation process exceeds the time of full dataset training solely under extreme threshold values. This divergence occurs when distilled dataset reaches equality with the full dataset in size post-distillation. Conversely, for pragmatic threshold values, the dataset distillation procedure consistently manifests as a significantly faster option to full dataset training.

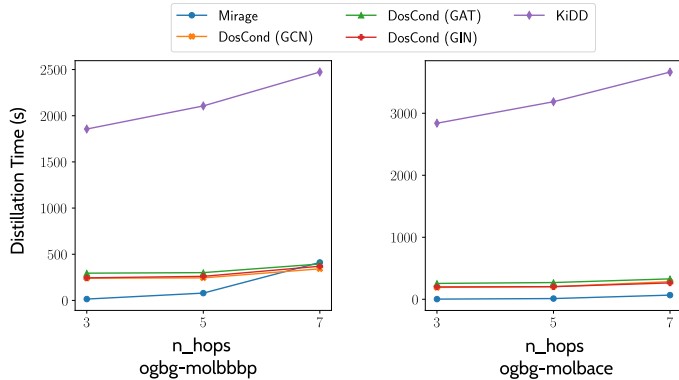

Figure H: Distillation times of MIRAGE, DOSCOND, and KIDD against the number of hops (layers) in the GNN.

Table I: Training time (in seconds) for full dataset

| Model → Dataset ↓ | GAT | GCN | GIN |
|---|---|---|---|
| ogbg-molbace | 98.08 | 73.49 | 72.99 |
| NCI1 | 90.71 | 145.09 | 120.96 |
| ogbg-molbbbp | 150.52 | 114.55 | 106.71 |
| ogbg-molhiv | 2744.21 | 1510.36 | 2418.61 |
| DD | 110.06 | 29.35 | 106.64 |
| IMDB-B | 12.84 | 11.42 | 9.81 |

## G   SUFFICIENCY OF FREQUENT TREE PATTERNS

This section contains the extended result of the experiment described in section 4.4 as shown in Fig. I. From the Fig. I, it is clearly visible that the dataset distilled using MIRAGE is able to capture the important information present in the full dataset since the difference between the losses when the full dataset is passed through the model and when the distilled dataset is passed through the model quickly approaches 0. This trend is held across models even though any information from the model was not used to compute the distilled dataset.

## H   PARAMETER VARIATIONS

In investigating the influence of the number of hops on the Area Under the Curve (AUC), we present a graphical representation of the AUC's variation in relation to the number of hops (Fig. K). Also, we depict the AUC in correlation with dataset sizes (Fig. L). It is important to note that sizes are closely tied to threshold parameters; however, the latter is not explicitly shown in the graphical representation due to their inherent high correlation with dataset sizes.

We see mild deterioration in AUC at higher hops in Fig. K. This is consistent with the literature since GNNs are known to suffer from oversmoothing and oversquashing at higher layers (Shirzad et al., 2023).

**Training Efficiency.** We now investigate the reduction in training loss over the course of multiple epochs. The outcomes for the datasets ogbg-molbace, ogbg-mohiv, DD, and IMDB-B are displayed in Fig. J. We selected these four datasets due to their representation of the smallest and largest graph dataset, the dataset with the largest graphs, and the densest graphs, respectively. Across all these datasets, the loss in the distilled dataset remains close to the loss in the full dataset. More interestingly, in three out of four datasets (DD and IMDB-B), the loss begins at a substantially lower value in the distilled dataset and approaches the minima quicker than in the full dataset. This trend provides evidence of MIRAGE's ability to achieve a dual objective. By identifying frequently co-

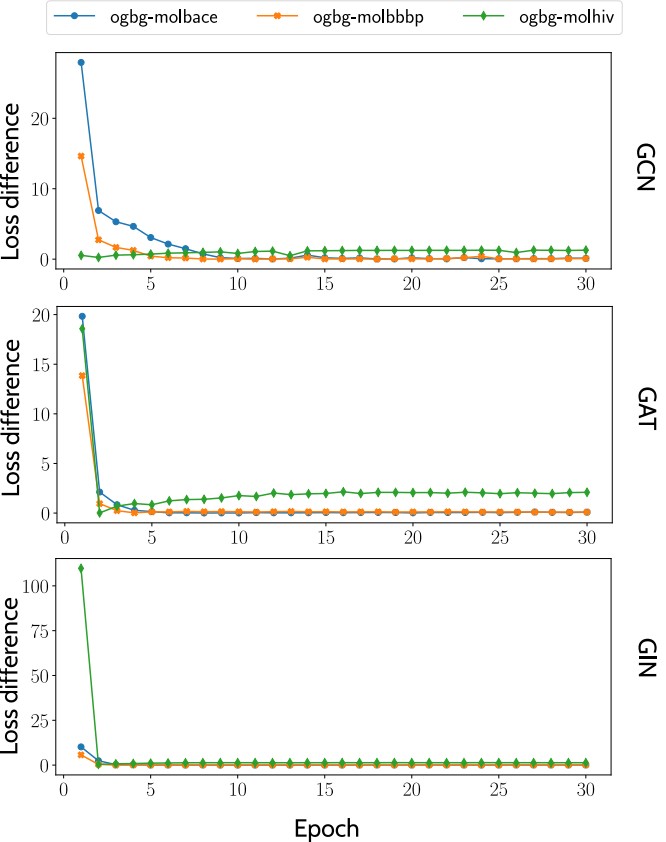

Figure I: Sufficiency of Frequent Tree Patterns: It is seen that the trend that the loss difference quickly approaches 0 holds across model and datasets.

(a) ogbg-molhiv      (b) DD      (c) IMDB-B      (d) ogbg-molbace

Figure J: Variation of training loss against the number of epochs in GCN.

occurring computation trees, we simultaneously preserve the most informative patterns within the dataset while effectively removing noise.

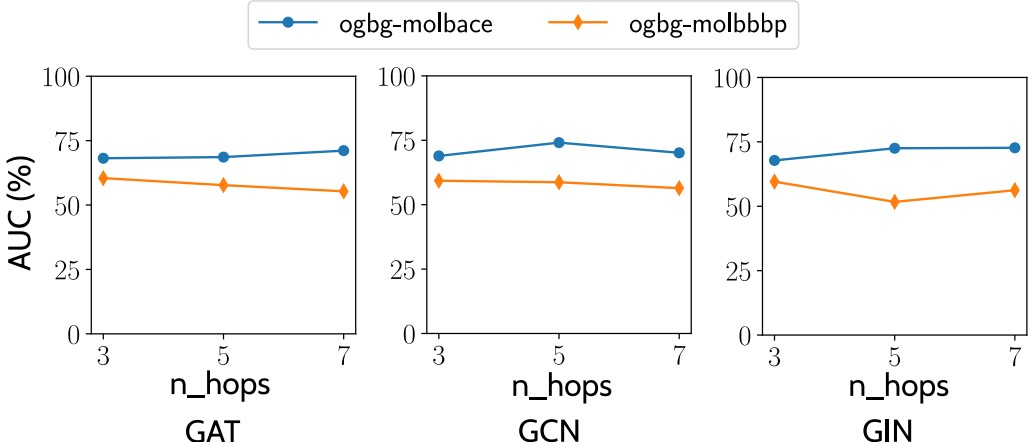

Figure K: ROC-AUC versus the n_hops parameter used during dataset distillation.

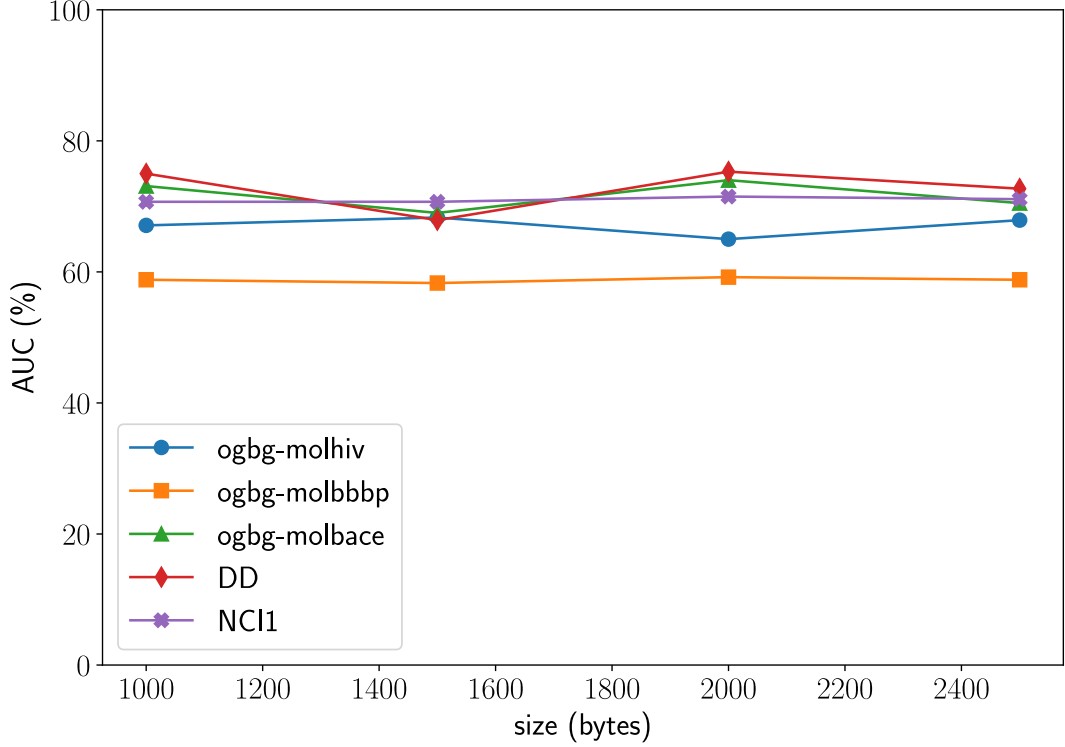

Figure L: Size vs. AUC for MIRAGE. Note that size is correlated to $\{\theta_0, \theta_1\}$

