# OpenReview forum: "Mirage: Model-agnostic Graph Distillation for Graph Classification"
_ICLR.cc/2024/Conference — ICLR 2024 poster_

### Official Review · Reviewer_XL6A · 2023-10-31

**Soundness:** 3 good
**Presentation:** 3 good
**Contribution:** 2 fair
**Rating:** 6
**Confidence:** 3

**Summary:**

This paper introduces a graph classification model called MIRAGE, which is a graph distillation method based on frequent pattern mining. MIRAGE takes advantage of the design features of the message passing framework, decomposes the graph into calculation trees, and uses the distribution characteristics of the calculation tree to compress calculation data. Compared with existing graph distillation algorithms, MIRAGE shows advantages in key indicators such as prediction accuracy, distillation efficiency, and data compression rate. Furthermore, MIRAGE only relies on CPU operations, providing a more environmentally friendly and energy-efficient graph distillation method.

**Strengths:**

1. It is a novel idea to transform the graph distillation problem into the problem of graph decomposition.
2. This paper clearly expresses the distillation problem in mathematical language and naturally transforms it into a graph decomposition problem.
3. The experimental configuration is described in detail, and the experimental results are good and reasonable.

**Weaknesses:**

1. The datasets used in the experiment is too small.

**Questions:**

1. Would MIRAGE still be as effective if used on a very large data set?
2. Why is MIRAGE, which only uses the CPU, so much faster than DOSCOND and KIDD, which can use the GPU?
3. If a datasets is large, but its global computation tree is very similar, will such a datasets affect the effect of MIRAGE?
4. How will MIRAGE perform in multi-classification tasks?

---

> ### Author Response · Authors · 2023-11-18
> **Response to Reviewer XL6A**
>
> **Q1. Would MIRAGE still be as effective if used on a very large dataset?**
>
> *Response:* We do not expect graph size to have any effect on the performance of MIRAGE since the underlying principles are not dependent on graph dataset size.  This is already evident from Table 2. Specifically, we compute the pearson's correlation coefficient between dataset size and the generalization error (AUC in full dataset-AUC of MIRAGE). The correlation is 0.153 with $p$-value of 0.743 and thus rejecting any bias in AUC from dataset size.
>
> **Q2. Why is MIRAGE, which only uses the CPU, so much faster than DOSCOND and KIDD, which can use the GPU?**
>
> *Response:* Existing algorithms, namely DosCOND and KiDD, seek to replicate the gradient trajectory of model parameters on the full dataset. Hence, they need to train on the full dataset for distillation (and as we comment in Sec 1, this compromizes the basic premise of data distillation). In contrast, MIRAGE does not require training on the full dataset. Rather, MIRAGE emulates the input data processed by message-passing GNNs. By shifting the computation task to the pre-learning phase, MIRAGE is not only faster, but also independent of the GNN architecture.
>
> **Q3. If a datasets is large, but its global computation tree is very similar, will such a datasets affect the effect of MIRAGE?**
>
> **Response:** As we discussed in our response to Q1, the size of the dataset does not affect the performance of MIRAGE. The performance depends primarily on the frequency distribution of computation trees in a dataset. If the distribution is skewed, MIRAGE is expected to perform well since the top-$k$ most frequent computation trees would effectively capture a substantial portion of the distribution mass. On the other hand, if the distribution is not skewed, MIRAGE is not the most suitable distillation technique for such scenarios. This aspect is explicitly acknowledged in our conclusion (Sec 5). The relevant text is reproduced verbatim below.
>
> > Finally, MIRAGE relies on the assumption that the distribution of computation trees is skewed. Although we provide compelling evidence of its prevalence across a diverse range of datasets, this assumption may not hold universally, especially in the case of heterophilous datasets. The development of a model-agnostic distillation algorithm remains an open challenge in such scenarios.
>
> **Q4. How will MIRAGE perform in multi-classification tasks?**
>
> *Response:* We have added a new dataset, namely IMDB-M, to evaluate MIRAGE's performance on multi-class classification. As we see below, MIRAGE produces the best performance. These results, and the details of the IMDB-M dataset has been added to Table 2 and Table 1 respectively in the revised manuscript.
>
> Architecture|RANDOM(Mean)|RANDOM(sum)|HERDING|Kidd|DosCond|Mirage|\|Full Dataset|
> -|-|-|-|-|-|-|-|
> GCN|55.10$\pm$ 3.8|52.90 $\pm$ 2.52|61.0 $\pm$ 2.4|57.1 $\pm$ 1.11|55.9 $\pm$ 1.06|**63.20 $\pm$ 1.12**|\|64.10 $\pm$ 1.1|
> GIN|60.1$\pm$ 2.67|56.30 $\pm 5.5$|58.47 $\pm$ 4.12|54.18 $\pm$ 0.90|58.30 $\pm$ 1.70|**61.8 $\pm$ 1.51**|\|64.8 $\pm$ 1.10 |
>
> The size of the dataset (in bytes) is as follows:
>
> | $\frac{\text{ Method}\rightarrow}{\text{ Dataset}\downarrow}$ | Herding(GCN) | Herding(GIN) | KiDD |DosCond(GCN) | DosCond(GIN) | MIRAGE |\|Full Dataset|
> |-------------------------------------------------------------------------|--------------|---------|---------|---------|------------|--------------|----------
> |IMDB-M|1,156|1,256|936|720|824|**228**|\|645,160|
> -----

---

> > ### Author Response · Authors · 2023-11-21
> > **Eagerly awaiting feedback**
> >
> > Dear Reviewer,
> >
> > Since we are only a day away from the completion of the discussion phase, we are eagerly awaiting your feedback on the revised manuscript.
> >
> > We have incorporated all of the suggestions made in the review including new experiments on distillation for multi-class classification. We would love to discuss more if any concern remains unaddressed. Otherwise, we would really appreciate it if you could support the paper by increasing the score.
> >
> > regards,
> >
> > Authors

---

> > > ### Author Response · Authors · 2023-11-22
> > > **Keenly seeking feedback from Reviewer XL6A**
> > >
> > > Dear Reviewer XL6A,
> > >
> > > As the author-reviewer discussion phase is nearing its conclusion, we would like to inquire if there are any remaining concerns or areas that may need further clarification. Your support during this final phase, especially if you find the revisions satisfactory, would be of great significance. Your feedback and evaluation hold a pivotal role in determining the ultimate fate of our work.
> > >
> > > Regards,
> > >
> > > Authors

---

> > > > ### Author Response · Authors · 2023-11-23
> > > > **Eagerly waiting feedback from Reviewer XL6A**
> > > >
> > > > Dear Reviewer XL6A,
> > > >
> > > > Thank you for your insightful comments.
> > > >
> > > > We have incorporated all suggestions, including additional experiments on distillation for multi-class classification.
> > > >
> > > > As the author-reviewer discussion phase is nearing its conclusion, we would like to politely enquire if you have any outstanding concerns regarding our work.
> > > >
> > > > As all other reviewers are leaning towards acceptance, your feedback is indeed important for determining the ultimate fate of our work.
> > > >
> > > > Sincerely
> > > >
> > > > Authors

---

### Official Review · Reviewer_yzvf · 2023-11-01

**Soundness:** 3 good
**Presentation:** 3 good
**Contribution:** 3 good
**Rating:** 6
**Confidence:** 4

**Summary:**

This paper presents a model-agnostic graph distillation algorithm for graph classification, which aims to distill the original graph dataset into a small and compact synthetic dataset maintaining competitive model performance. It is pointed out that existing distillation methods typically rely on model-related information and need to train on the full dataset, resulting in poor generalization and efficiency issues, respectively. To address these weaknesses, MIRAGE directly compresses the computation trees to synthesize datasets by mining sets of frequently co-occurring items. This is inspired by the observation that the distribution of the computation trees, formed by decomposing graphs based message-passing GNN frameworks, always exhibits high skewness. Therefore, it is possible to approximate the true graph representation by aggregating the root representations of only the highly frequent trees. Extensive benchmarking demonstrates the superiority of MIRAGE.

**Strengths:**

1. This paper distills the original graph dataset into a novel form, namely the sets of computation trees, instead of traditional feature and structure information, i.e. X’ and A’.

2. This paper not only improves generalization by avoiding model-related computation but also addresses the limitation of the traditional distillation step that necessitates training on the entire dataset.

3. MIRAGE demonstrates superiority from various metrics, i.e. predictive accuracy, distillation efficiency, and data compression rates.

**Weaknesses:**

1. Unlike DOSCOND and KIDD, MIRAGE seems infeasible to be generalized to node classification tasks.

2. There is no theoretical analysis nor experimental proof to investigate the error when approximating the true graph representation by aggregating the root representations of only the highly frequent trees.

3. MIRAGE compresses the computation trees by mining the set of frequently co-occurring items. However, we can consider an extreme situation in which two computation trees are co-occurring in all graphs, but their frequencies in each graph are very low. So, these trees can’t approximate the true graph representation, which will destroy the synthetic dataset performance.

4. The synthetic dataset can solely be used to train GNNs based on the message-passing framework, but can’t used to train certain spectral GNNs, such as ChebyNet with multiple propagation steps.

**Questions:**

1. Original dataset labels are still used during the distillation procedure. So is it a strictly unsupervised algorithm from this perspective?

2. There is no clear statement about how Figure 1 is plotted. And I can't get what the x-axis and y-axis represent, respectively.

---

> ### Author Response · Authors · 2023-11-18
> **Response to Reviewer yzvf: Part 1**
>
> **1. Unlike DOSCOND and KIDD, MIRAGE seems infeasible to be generalized to node classification tasks.**
>
> *Response:* While the primary focus of MIRAGE is on graph classification (as noted in the title itself), MIRAGE does generalize to node classification. We now empirically demonstrate this ability in App. E. (*Note:* KIDD also does not discuss node classification. Only, DosCOND does).
>
> Generalization to node classification is simple since learning the node embeddings is a necessary step before aggregating them to form the graph embedding (Recollect Eqs. 4 and 5). Specifically, the pipeline follows in an identical manner except the omission of (Eq. 5) and the loss function being applied to node embeddings instead of graph embeddings.
>
> We use ogbg-molbace to analyze performance in node classification. Here, each node is labeled as aromatic or non-aromatic depending on whether it is part of an aromatic ring substructure. The results are tabulated in Table H. Consistent with previous results, MIRAGE outperforms DosCOND in both AUC-ROC and compression (smaller size of the distilled dataset). DosCOND produces a dataset that is $\approx 4$ times the size of that produced by MIRAGE, yet performs more than $7\%$ lower in AUC-ROC. This coupled with model-agnostic-ness further solidifies the superiority of MIRAGE
>
> **2. There is no theoretical analysis nor experimental proof to investigate the error when approximating the true graph representation by aggregating the root representations of only the highly frequent trees.**
>
> *Response:* The above question is indeed important. We have now included a new experiment to evaluate the *sufficiency* of the frequent computation tree patterns.
>
>  **Sufficiency of frequent patterns:** In order to establish sufficiency of frequent tree patterns in capturing the dataset characteristic, we conduct the following experiment. We train the model on the full dataset and store its weights at each epoch. Then, we freeze the model at the weights after each epoch's training and pass both the distilled dataset consisting of just the frequent tree patterns, and the full dataset. We then compute the differences between the losses as shown in Figure I (GCN, GAT, and GIN). The rationale behind this is that the weights of the full model recognise the patterns that are important towards minimizing the loss. Now, if the same weights continue to be effective on the distilled train set, it indicates that the distilled dataset has retained the important information. In the figure, we can see that the difference quickly approaches $0$ for all the models for all the datasets, and only starts at a high value at the random initialization where the weights are not yet trained to recognize the important patterns. Furthermore, gradient descent will run more iterations on trees that it sees more often and hence infrequent trees have limited impact on the gradients. These results empirically establish the sufficiency of frequent tree patterns in capturing the majority of the dataset characteristic.
>
>  **Additional information relevant to this question:** In Fig. 1, we show that the distribution of computational trees is long-tailed and hence by considering only the top-$k$ most frequent trees, majority of the input information processed by the GNN would be retained. Furthermore, in Fig. 5 (b), we show that the training loss against epochs is similar for the full dataset and distilled dataset.

---

> > ### Author Response · Authors · 2023-11-18
> > **part 2**
> >
> > **3. MIRAGE compresses the computation trees by mining the set of frequently co-occurring items. However, we can consider an extreme situation in which two computation trees are co-occurring in all graphs, but their frequencies in each graph are very low. So, these trees can’t approximate the true graph representation, which will destroy the synthetic dataset performance.**
> >
> > **Response:**  The observation is correct that if the frequencies of the co-occurring pair within all graphs are consistently low (and no other frequent tree sets exist across these graphs), then the approximated graph embedding may be different from the true graph embedding. However, it is essential to clarify that the performance on the synthetic dataset is not contingent on the distance between the approximated and true graph embeddings. Instead, it hinges on whether the approximated embedding includes the necessary information for predicting the class label of the graphs.
> >
> > Given this context, recollect that we mine frequent tree sets per class label (Section 3.4). Therefore, if the residual tree sets across these graphs exhibit diversity, with the co-occurring pair being the sole exception, it becomes challenging for the GNN to establish correlations from the residual portion not reflected in the approximated embedding. Consequently, the embedding constructed solely from the shared computation trees across graphs may still retain valuable information.
> >
> > Nonetheless, we realize that there may be situations such as the one described here, or when the distribution of computation trees is not skewed, MIRAGE may not be the most suitable method. We now explicitly acknowledge this aspect in our conclusion. We present below the relevant text verbatim.
> >
> > > MIRAGE relies on the assumption that the distribution of computation trees is skewed. Although we provide compelling evidence of its prevalence across a diverse range of datasets, this assumption may not hold universally, especially in the case of heterophilous datasets. The development of an unsupervised, model-agnostic distillation algorithm remains an open challenge in such scenarios.
> >
> > **4. The synthetic dataset can solely be used to train GNNs based on the message-passing framework, but can’t used to train certain spectral GNNs, such as ChebyNet with multiple propagation steps.**
> >
> > **Response:** We agree and this aspect was already acknowledged in our Conclusion. We present below the exact text that attests to this limitation. We note that this limitation also exists for KIDD since it assumes the input GNN architecture can be equivalently expressed as a Graph Neural Tangent Kernel, which holds for MPNNs but not graph transformers or spectral GNNs.
> >
> > > Additionally, there is a need to assess how these existing algorithms perform on contemporary architectures like graph transformers (e.g., (Ying et al., 2021; Ramp ́aˇsek et al., 2022)) or equivariant GNNs (e.g., (Satorras et al., 2022)). Our future work will be dedicated to exploring these avenues of research.
> >
> >
> > **5. Original dataset labels are still used during the distillation procedure. So is it a strictly unsupervised algorithm from this perspective?**
> >
> > **Response:** We thank the reviewer for highlighting this subtle point. We call our algorithm unsupervised from the perspective of not using the training gradients on the full dataset. To address this issue, we do not use the word "unsupervised" in isolation anywhere in the revised version. In all places, we mention "unsupervised to original training gradients".
> >
> > **6. There is no clear statement about how Figure 1 is plotted. And I can't get what the x-axis and y-axis represent, respectively.**
> >
> > **Response:** We apologize for not being clear in our description. We have now incorporated a detailed figure caption, which is reproduced verbatim below.
> >
> > > The figure presents the frequency distribution of computation trees across datasets. The "frequency" of a computation tree denotes the number of occurrences of that specific tree across all graphs in a dataset. The *normalized* frequency of a tree is computed by dividing its frequency with the total number of trees in a dataset and thus falls in the range $[0,1]$. The $x$-axis of the plot depicts the normalized frequency counts observed in a dataset, while the $y$-axis represents the percentage of computation trees corresponding to each frequency count. Both $x$ and $y$ axes are in log scale. The distribution is highly skewed characterized by a dominance of trees with low frequency counts, while a small subset of trees exhibiting higher frequencies. For example, in ogbg-molhiv, the most frequent tree alone has normalized frequency of $0.32$.

---

> > > ### Comment · Reviewer_yzvf · 2023-11-21
> > >
> > > Thanks for the clarification and I will maintain my score.

---

### Official Review · Reviewer_pdUR · 2023-11-01

**Soundness:** 3 good
**Presentation:** 3 good
**Contribution:** 4 excellent
**Rating:** 6
**Confidence:** 4

**Summary:**

This paper studies a new graph distillation approach called MIRAGE for graph classification, which addresses limitations in existing methods. MIRAGE leverages the idea that message-passing GNNs break down input graphs into computation trees with skewed frequency distributions, allowing for concise data summarization. Unlike traditional methods that emulate gradient flows, MIRAGE compresses the computation data itself, making it an unsupervised and architecture-agnostic distillation algorithm.

**Strengths:**

1. It is crucial to address the issue of existing costs in training large-scale graphs through graph distillation, as it remains an under-explored problem.
2. It is interesting to observe how the authors have proposed an unsupervised approach for graph distillation, eliminating the requirement for parameters and model architecture, and still achieving comparable performance to the original dataset.
3. The methodology is interesting, and the performances achieved by their approach are amazing.

**Weaknesses:**

1. In the results presented in Table 2, why there is a discrepancy compared to those from the original DosCond paper? Could the authors clarify why the authors deviated from the settings employed in the DosCond paper?

2. It would be beneficial if the authors could delve deeper into the sensitivity of their approach with respect to distillation parameters \theta_1 and \theta_2. Undertaking further experiments to report the performance across varied distillation parameters might offer greater insights, given that these parameters significantly influence the size of the distilled dataset.

3. On the ogbg-molhiv dataset, could the authors provide an explanation for the better performance observed achieved by distilled data compared to the original dataset.

4. As we are removing the nodes that exist in the less-occurring computational trees, how can we be sure that we are not losing important information in the dataset (especially since this process does not consider the feature information)?  Can authors justify it theoretically or/and empirically by running some experiments?

**Questions:**

My major questions are on the experimental evaluation. It would tremendously strengthen this work by addressing the concerns listed in the Weakness section.

---

> ### Author Response · Authors · 2023-11-18
> **Response to Reviewer pdUR: part 1**
>
> **Q1. In the results presented in Table 2, why there is a discrepancy compared to those from the original DosCOND paper? Could the authors clarify why the authors deviated from the settings employed in the DosCOND paper?**
>
> *Response:* The results in DosCOND do not exactly align with ours due to the following reasons:
> * **Input data:** The graph datasets used (both in ours and DosCOND) are node and edge labeled. In DosCOND, ***only*** the node labels are used for predictive modeling, whereas we use both node and edge labels. We believe this is a better empirical design since generalization error between full and distilled dataset should be measured after incorporating the full input information.
> * **Metrics:** A direct comparison between the AUCROC numbers of DosCOND and ours is possible only in the three datasets of ogbg-molbace, ogbg-mohiv, and ogbg-molbbbp, since DosCOND does not use AUCROC as the evaluation metric in the other datasets. We use AUCROC across all datasets.
>     * **Performance of DosCOND:** In all three datasets, as we show in the table below (columns 2 and 3), **DosCOND's AUCROC improves in our evaluation.**
>     * **AUCROC on full dataset training**: We observe significantly reduced AUCROC only in ogbg-molhiv (columns 4 and 5). We conducted hyper-parameter search over a larger space than in our initial setup for ogbg-molhiv (full dataset training) to revisit this issue and identified that reducing the weight of the L2 regularizer in the loss function from 0.01 to 0.00001 improves AUCROC to $75.93$ in GCN. A similar improvement is observed in GAT and GIN as well. This update has now been incorporated in the revised draft (highlighted in blue font in Table 2).
>
> Dataset|Doscond AUCROC in DosCond (#Graphs=1)| DosCond AUCROC in ours (GCN) | Full dataset AUCROC in DosCond | Full Dataset AUCROC in ours (GCN)
> -|-|-|-|-|
> ogbg-molbace|65.7|**67.34**|71.4|**77.31**
> ogbg-molbbbp|58.1|**61.30**|64.6|64.43
> ogbg-molhiv|72.6|**73.16**|**75.7**|64.78 $\rightarrow$ 75.93
>
> * **Other things to note:** DosCond reports numbers only for GCN, whereas we include GCN, GAT and GIN. The budget used in our experiments is set to 1 graph per class since even with this budget, MIRAGE is smaller than DosCond (details in Table 3).
>
> **Q2. It would be beneficial if the authors could delve deeper into the sensitivity of their approach with respect to distillation parameters $\theta_1$ and $\theta_2$. Undertaking further experiments to report the performance across varied distillation parameters might offer greater insights, given that these parameters significantly influence the size of the distilled dataset.**
>
> *Response:* We appreciate this constructive feedback. We have significantly expanded the studies on parameter sensitivity. Below we summarize the experiments conducted and the key insights derived from them.
>
> **Impact of Frequency Threshold on distillation time, compression and accuracy (AUCROC):** In Section 4.5 (Fig. 6 presents the results), we measure the impact of frequency threshold on the distillation time. MIRAGE is significantly faster than training on full datasets unless the frequency thresholds are set to extreme values where the distilled dataset reaches equality with the full dataset.
>
> In Fig. L in Appendix, we present the relationship between compression and AUCROC. The compression ratio is controlled by increasing the frequency threshold, i.e., the higher the threshold, the more the compression is. As noted, there is marginal improvement in AUCROC with lower thresholds (higher distillation size). This trend can be explained from the long-tailed distribution of computation trees. Even if we incorporate more computation trees, they have negligible impact on capturing more of the distribution mass and hence limited effect on AUCROC.
>
> **Impact of number of hops (layers):**  The details are furnished in Appendix F and H of the appendix. We note the main observations here. First, even under high hop counts from the GNN perspective, the distillation process is more time-efficient than complete dataset training. Moreover, the distilled dataset's performance converges closely with that of the full dataset, as evident in Table 2. Second, the distillation process exceeds the time of full dataset training solely under extreme threshold values. This divergence occurs when distilled dataset reaches equality with the full dataset in size post-distillation.
>
> In Fig. K in Appendix, we present the impact of hops on the AUC. We see mild deterioration in AUC at higher hops. This is consistent with the literature since GNNs are known to suffer from oversmoothing and oversquashing at higher layers[1].
>
> In summary, for pragmatic threshold values, the dataset distillation procedure consistently manifests as a significantly faster option to full dataset training.
>
> [1] Hamed Shirzad, Ameya Velingker, Balaji Venkatachalam, Danica J. Sutherland, and Ali Kemal Sinop. Exphormer: Sparse transformers for graphs. In ICML, 2023

---

> > ### Author Response · Authors · 2023-11-18
> > **part 2**
> >
> > **Q3. On the ogbg-molhiv dataset, could the authors provide an explanation for the better performance observed achieved by distilled data compared to the original dataset.**
> >
> > *Response:* As noted in our response to Q1, the performance on distilled dataset of ogbg-mohiv no longer outperforms the full dataset. Nonetheless, as we note in our manuscript, improved performance on the distilled dataset is possible and has been observed in the literature (Ex. In KiDD also this phenomenon has been reported). We reproduce the relevant text verbatim below.
> >
> > > We observe instances, such as in DD, where the distilled dataset outperforms the full dataset, an outcome that might initially seem counter-intuitive. This phenomenon has been reported in the literature before~\citep{KIDD}. While pinpointing the exact cause behind this behavior is challenging, we hypothesize that the distillation process may tend to remove outliers from the training set, subsequently leading to improved accuracy. Additionally, given that distillation prioritizes the selection of graph components that are more informative to the task, it is likely to retain the most critical patterns, resulting in enhanced model generalizability.
> >
> > **Q4. As we are removing the nodes that exist in the less-occurring computational trees, how can we be sure that we are not losing important information in the dataset (especially since this process does not consider the feature information)? Can authors justify it theoretically or/and empirically by running some experiments?**
> >
> >  *Response:* The above question is indeed important. We have now included a new experiment to evaluate the *sufficiency* of the frequent computation tree patterns.
> >
> >  **Sufficiency of frequent patterns:** In order to establish sufficiency of frequent tree patterns in capturing the dataset characteristic, we conduct the following experiment. We train the model on the full dataset and store its weights at each epoch. Then, we freeze the model at the weights after each epoch's training and pass both the distilled dataset consisting of just the frequent tree patterns, and the full dataset. We then compute the differences between the losses as shown in Figure I (GCN, GAT, and GIN). The rationale behind this is that the weights of the full model recognise the patterns that are important towards minimizing the loss. Now, if the same weights continue to be effective on the distilled train set, it indicates that the distilled dataset has retained the important information. In the figure, we can see that the difference quickly approaches $0$ for all the models for all the datasets, and only starts at a high value at the random initialization where the weights are not yet trained to recognize the important patterns. Furthermore, gradient descent will run more iterations on trees that it sees more often and hence infrequent trees have limited impact on the gradients. These results empirically establish the sufficiency of frequent tree patterns in capturing the majority of the dataset characteristic.
> >
> > **Feature information:** Finally, we wanted to point out that we do consider feature information. In Def 2, we define graph (tree) isomorphism where feature correspondence is a required condition (Constraint (2)). Frequency is counted based on this definition (Eq. 6).
> >
> >  **Additional information relevant to this question:** In Fig. 1, we show that the distribution of computational trees is long-tailed and hence by considering only the top-$k$ most frequent trees, majority of the input information processed by the GNN would be retained. Furthermore, in Fig. 5 (b), we show that the training loss against epochs is similar for the full dataset and distilled dataset.

---

> > > ### Author Response · Authors · 2023-11-21
> > > **Eagerly awaiting feedback on rebuttal**
> > >
> > > Dear Reviewer,
> > >
> > > Since we are only a day away from the completion of the discussion phase, we are eagerly awaiting your feedback on the revised manuscript.
> > >
> > > Your review pointed out important empirical studies that further enhanced our work. We have incorporated all of them and we thank the reviewer again for the deep insightful comments on our work. We would love to discuss more if any concern remains unaddressed. Otherwise, we would really appreciate it if you could support the paper by increasing the score.
> > >
> > > regards,
> > >
> > > Authors

---

> > > > ### Comment · Reviewer_pdUR · 2023-11-21
> > > > **Thanks for the rebuttal**
> > > >
> > > > Thanks for the response. I tend to accept this paper and will maintain my score.

---

### Official Review · Reviewer_2KBE · 2023-11-01

**Soundness:** 3 good
**Presentation:** 3 good
**Contribution:** 2 fair
**Rating:** 6
**Confidence:** 3

**Summary:**

This paper proposes MIRAGE, a model-agnostic graph distillation method for graph classification task. Specifically, the authors first observe that existing graph distillation methods still rely on training with the full dataset and specific to GNNs architectures and hyper-parameters. To this end, MIRAGE can tackle these two limitations via decomposing input graphs into computation trees. The long-tailed frequency distribution of the computation tree enables the graph distillation with a high compression ratio. Experiments demonstrate the superiority of MIRAGE in terms of its effectiveness, compression ratio, and distillation efficiency.

**Strengths:**

1.	This paper is well-organized and easy to follow.
2.	The idea of making graph distillation model-agnostic is interesting and meaningful in practice.
3.	The proposed data-centric solution is a natural way to mitigate model hyperparameter dependency using the frequency distribution of the computation tree.

**Weaknesses:**

1. Motivations:
- (a) Problem Motivation: In the abstract, one of the identified limitations pertains to the reliance on training with the full dataset. To provide a clearer argument regarding the necessity of full dataset training for existing methods and the distinct advantages of MIRAGE, we should consider the following: What is the comparative performance of existing distillation methods (e.g., gradient matching) when applied to both randomly initialized and well-trained models? Is there a substantial performance gap between the two scenarios? Regarding computational costs in the distillation process, it's worth examining the resource requirements for calculating the computation tree frequency distribution, particularly in the context of deep GNNs. How does this computational overhead compare to training models on the full dataset?
- (b) Technical Motivation: While the concept of the computation tree is inherent to GNNs, we should address the sufficiency of the frequency distribution for the computation tree. Key points to explore include: Why is maintaining a high computation tree frequency distribution sufficient for effective learning? What is the distribution pattern of the computation tree, and does it exhibit a long-tail distribution?

2. Objectives: Ideally, the distillate data should mirror the distribution of the full dataset, implying that the gradients of the distillate data should align with those of the full dataset for any model (random or well-trained). It's essential to clarify that while distillation can reduce computation costs, the primary objective remains data distillation. Additionally, considering the architecture transferability for existing graph distillation methods, the advantage of model-agnostic techniques is not significant.

3. Experiments:
- (a) Comprehensive Model Comparison: In evaluating baselines, it would be beneficial to perform a comprehensive comparison across various model architectures. Specifically, assessing performance across different architecture pairs would enhance the evaluation's robustness and provide insights into transferability.

- (b) Baselines and Model Initialization: Investigate the significance of GNN model training prior to distillation for baselines. Explore performance outcomes for well-trained, randomly initialized, and warm-up trained models. Understanding the effectiveness of random initialization can help determine the necessity of training on the full data for baseline methods.

- (c) Hyperparameter Investigation: Explore the impact of different hyperparameters, such as threshold values and the number of hops, on various metrics, including accuracy, efficiency, compression ratio, and transferability.

- (d) Trade-off Analysis: Provide insights into the trade-off between accuracy, efficiency, and compression ratio. Understanding these trade-offs can offer valuable guidance for practical applications.

In a nutshell, I am currently leaning toward weak rejection. I am looking forward to the authors’ response.

-------------------After rebuttal -------------------------
Thanks for the response. I increase my score to 6.

**Questions:**

Please see the weakness part.

---

> ### Author Response · Authors · 2023-11-18
> **Response to Reviewer 2KBE: Part 1**
>
> **1a(i). Problem motivation: What is the comparative performance of existing distillation methods (e.g., gradient matching) when applied to both randomly initialized and well-trained models? Is there a substantial performance gap between the two scenarios?**
>
> *Response:* Below we present the AUCROC numbers of DosCOND on randomly initialized DosCOND model, warm start (100 epochs) and training DosCOND until convergence (typically 1000 epochs). As visible, there is a noticeable gap in the AUCROC numbers indicating training until convergence is necessary. This discussion has been added App. D.
>
> |Dataset|Random|Warm start|Convergence|
> |-|-|-|-|
> |ogbg-molbace|$55.04\pm9.07$|$59.25\pm1.61$|$67.34\pm1.84$|
> |NCI1 | $51.22\pm2.00$|$48.18\pm2.78$|$57.90\pm0.75$|
> |ogbg-molbbbp|$52.64\pm1.98$|$50.72\pm3.48$|$59.19\pm0.95$|
> |ogbg-molhiv|$48.21\pm5.95$|$34.99\pm7.25$|$73.16\pm0.69$|
> |DD|$52.39\pm7.19$|$61.58\pm2.11$|$68.39\pm9.64$|
>
>
> **1a(ii). Regarding computational costs in the distillation process, it's worth examining the resource requirements for calculating the computation tree frequency distribution, particularly in the context of deep GNNs. How does this computational overhead compare to training models on the full dataset?**
>
> *Response:* The time taken to distill each dataset is provided in Fig. 4(a) in the form of a bar plot and the raw numbers are in Table E. MIRAGE is $\approx$ 500 times faster on average than KIDD and $\approx 150$ times faster than DOSCOND. To further expand on this aspect, we have added the following new experiments:
> * **Comparison to full dataset training time:** We now report the time taken to train on the full dataset in Table I in Appendix. On average, the distillation time of MIRAGE is more than $30$ times faster than training on the full dataset. In contrast DosCOND and KiDD are slower than full dataset training due to reasons already highlighted in Sec 1 of our manuscript.
> * **Impact of hops (layers):** In Fig. H (discussed in App F and referred to from Sec 4.4 in main paper), we report the growth in distillation time against the number of hops. We show that even at a larger values of hops, MIRAGE is 2 to 25 times faster, on average, than training on the full dataset.
> * **Impact of frequency threshold on distillation:** we subject the system to variations in threshold parameters, graphically showing the resulting time in Figure 6. Notably, the distillation process exceeds the time of full dataset training solely under extreme threshold values. This divergence occurs when distilled dataset reaches equality with the full dataset in size post-distillation. Conversely, for pragmatic threshold values, the dataset distillation procedure consistently manifests as a significantly faster option to full dataset training.
>
>
> **1b. Technical Motivation: While the concept of the computation tree is inherent to GNNs, we should address the sufficiency of the frequency distribution for the computation tree. Key points to explore include: Why is maintaining a high computation tree frequency distribution sufficient for effective learning? What is the distribution pattern of the computation tree, and does it exhibit a long-tail distribution?**
>
>  *Response*: We note that the frequency distribution of computation trees is already reported in our manuscript in Fig. 1 and it is indeed a long-tail distribution. To further establish the sufficiency of frequent tree patterns, we have now dedicated an explicit section (Sec 4.4).
>
>  **Sufficiency of frequent patterns:** In order to establish sufficiency of frequent tree patterns in capturing the dataset characteristic, we conduct the following experiment. We train the model on the full dataset and store its weights at each epoch. Then, we freeze the model at the weights after each epoch's training and pass both the distilled dataset consisting of just the frequent tree patterns, and the full dataset. We then compute the differences between the losses as shown in Figure 5(a) (GCN) and Figure I (GCN, GAT, and GIN). The rationale behind this is that the weights of the full model recognise the patterns that are important towards minimizing the loss. Now, if the same weights continue to be effective on the distilled train set, it indicates that the distilled dataset has retained the important information. In the figure, we can see that the difference quickly approaches $0$ for all the models for all the datasets, and only starts at a high value at the random initialization where the weights are not yet trained to recognize the important patterns. Furthermore, gradient descent will run more iterations on trees that it sees more often and hence infrequent trees have limited impact on the gradients. These results empirically establish the sufficiency of frequent tree patterns in capturing the majority of the dataset characteristics.

---

> ### Author Response · Authors · 2023-11-18
> **part 2**
>
> **2(i). Objectives: Ideally, the distillate data should mirror the distribution of the full dataset, implying that the gradients of the distillate data should align with those of the full dataset for any model (random or well-trained). It's essential to clarify that while distillation can reduce computation costs, the primary objective remains data distillation.**
>
> *Response*: As mentioned in our response to 1b, the frequency distribution of computation trees is long tailed. With the newly added sufficiency experiments, we now also show that the frequent trees are the patterns that are being learned by the model parameters as well (Response to 1b.). Finally in Fig. 5b and J, we also show that the train losses on full dataset and distilled datasets, with their own respective training weights, are also aligned.
>
> **2(ii). Objectives: Additionally, considering the architecture transferability for existing graph distillation methods, the advantage of model-agnostic techniques is not significant.**
>
> *Response:* Our manuscript includes experiments to evaluate transferability of model architectures in graph distillation. The results, reported in Table F of Appendix C, reveal there is a noticeable drop in AUCROC when the model architecture for distillation is different than the architecture used for training on the distilled dataset. The same observation has been reported in another recent work as well (See Table 3 in [1]). In KiDD [2], they theoretically establish why the distillation GNN and the training GNN needs to be same (Page 3 in KiDD). These results provide verification of why model-agnostic distillation is a problem of importance.
>
> To better highlight these results and motivate the proposed problem, we now point to this information from the introduction itself.
>
> * [1] Beining Yang, Kai Wang, Qingyun Sun, Cheng Ji, Xingcheng Fu, Hao Tang, Yang You, and Jianxin Li. Does graph distillation see like vision dataset counterpart? NeurIPS, 2023*.(to appear)
> * [2] Zhe Xu, Yuzhong Chen, Menghai Pan, Huiyuan Chen, Mahashweta Das, Hao Yang, and Hanghang Tong. Kernel ridge regression-based graph dataset distillation. ACM SIGKDD, 2023
>
> **Experiments: 3a. Comprehensive Model Comparison: In evaluating baselines, it would be beneficial to perform a comprehensive comparison across various model architectures. Specifically, assessing **performance across different architecture pairs** would enhance the evaluation's robustness and provide insights into transferability.**
>
> *Response:* Our evaluation already includes performance assessment across three different GNN architectures namely- GCN, GAT, and GIN, in Table 2. The results establish the proposed technique to be superior on average and consistently ranked among the top-2 distillation algorithms across all dataset-architecture combinations.
>
> In addition, cross architecture transferability, i.e., the situation where the architecture used for distillation is different from the one used for training and inference is also included in Appendix C. As discussed in our response to 2(ii), there is a noticeable drop in AUCROC, which is consistent with existing literature [1]. This result motivates the need for model-agnostic distillation algorithms.
>
> **3b. Baselines and Model Initialization: Investigate the significance of GNN model training prior to distillation for baselines. Explore performance outcomes for well-trained, randomly initialized, and warm-up trained models. Understanding the effectiveness of random initialization can help determine the necessity of training on the full data for baseline methods.**
>
>  *Response*: Below we present the AUCROC numbers of DosCOND on randomly initialized DosCOND model, warm start (100 epochs) and training DosCOND until convergence (typically around 1000 epochs). As visible, there is a noticeable gap in the AUCROC numbers indicating full training is necessary. This discussion has been added App. D.
>
>
> |Dataset|Random|Warm start| Converged|
> |-|-|-|-|
> |ogbg-molbace|$55.04\pm9.07$|$59.25\pm1.61$|**67.34$\pm$ 1.84**|
> |NCI1 | $51.22\pm2.00$|$48.18\pm2.78$|**57.90$\pm$ 0.75**|
> |ogbg-molbbbp|$52.64\pm1.98$|$50.72\pm3.48$|**59.19$\pm$ 0.95**|
> |ogbg-molhiv|$48.21\pm5.95$|$34.99\pm7.25$|**73.16$\pm$ 0.69**|
> |DD|$52.39\pm7.19$|$61.58\pm2.11$|**68.39$\pm$ 9.64**|

---

> > ### Author Response · Authors · 2023-11-18
> > **part 3**
> >
> > **3c. Hyperparameter Investigation: Explore the impact of different hyperparameters, such as threshold values and the number of hops, on various metrics, including accuracy, efficiency, compression ratio, and transferability.**
> >
> > *Response:* All of these experiments have been incorporated in the revised manuscript.
> >
> > **Impact of Frequency Threshold on distillation time, compression and accuracy (AUCROC):** In Section 4.5 (Figure 6 presents the results), we measure the impact of frequency threshold on the distillation time. MIRAGE is significantly faster than training on full datasets unless the frequency thresholds are set to extreme values where the distilled dataset reaches equality with the full dataset.
> >
> > In Fig. L in Appendix, we present the relationship between compression and AUCROC. The compression ratio is controlled by increasing the frequency threshold, i.e., the higher the threshold, the more is the compression. As noted, there is marginal improvement in AUCROC with lower thresholds (higher distillation size). This trend can be explained from the long-tailed distribution of computation trees. Even if we incorporate more computation trees, they have negligible impact on capturing more of the distribution mass and hence limited effect on AUCROC.
> >
> > **Impact of number of hops (layers):**  The details are furnished in Appendix F and H of the appendix. We note the main observations here. First, even under high hop counts from the GNN perspective, the distillation process is more time-efficient than complete dataset training. Moreover, the distilled dataset's performance converges closely with that of the full dataset, as evident in Table 2. Second, the distillation process exceeds the time of full dataset training solely under extreme threshold values. This divergence occurs when distilled dataset reaches equality with the full dataset in size post-distillation.
> >
> > In Fig. K in Appendix, we present the impact of hops on the AUC. We see mild deterioration in AUC at higher hops. This is consistent with the literature since GNNs are known to suffer from oversmoothing and oversquashing at higher layers[1].
> >
> > **Transferability:** Our algorithm is model-agnostic. Hence, transferability of knowledge from one model to another is not relevant for MIRAGE.
> >
> > In summary, for pragmatic threshold values, the dataset distillation procedure consistently manifests as a significantly faster option to full dataset training.
> >
> > [1] Hamed Shirzad, Ameya Velingker, Balaji Venkatachalam, Danica J. Sutherland, and Ali Kemal Sinop. Exphormer: Sparse transformers for graphs. In ICML, 2023
> >
> > **3d. Trade-off Analysis: Provide insights into the trade-off between accuracy, efficiency, and compression ratio. Understanding these trade-offs can offer valuable guidance for practical applications.**
> >
> > *Response:* As mentioned in our response to Q3c above, the trade-off between compression ratio and AUCROC has been discussed in detail in App H (also references from Sec 4.5 in main paper). In our response to 3c., we also discuss the impact of frequency threshold on the distillation time. This is also discussed in detail in Sec 4.5.

---

> > > ### Author Response · Authors · 2023-11-20
> > > **Looking forward to your feedback**
> > >
> > > Dear Reviewer,
> > >
> > > We thank you for taking the time to provide constructive comments, which have significantly improved the quality of the manuscript. In our revision, we have incorporated all of the given suggestions including:
> > >
> > > * Establishing sufficiency of frequent patterns through detailed empirical studies.
> > > * Detailed study on the impact of random, warm-start and fully trained models on supervised distillation algorithms
> > > * Extensive study of hyper-parameter sensitivity on accuracy (AUCROC), compression and running time.
> > >
> > > With these additional experiments and improved explanations, we hope we have addressed all the concerns raised by the reviewer. If there are any outstanding concerns, we request the reviewer to please raise those. Otherwise, we would really appreciate it if the reviewer could increase the score.
> > >
> > > Looking forward to your response.
> > >
> > > Thank you,
> > >
> > > Authors

---

> > > > ### Author Response · Authors · 2023-11-21
> > > > **Eagerly waiting for post-rebuttal feedback**
> > > >
> > > > Dear Reviewer,
> > > >
> > > > We thank you for the insightful comments on our work. Your suggestions have now been incorporated in our revision and we are eagerly waiting for your feedback. As the author-reviewer discussion phase is approaching its conclusion in just a few hours, we are reaching out to inquire if there are any remaining concerns or points that require clarification. Your feedback is crucial to ensure the completeness and quality of our work.
> > > >
> > > > We are pleased to share that the responses from other reviewers indicate a positive inclination towards acceptance. Your support in this final phase, particularly if you find the revisions satisfactory, would be immensely appreciated.
> > > >
> > > > regards,
> > > >
> > > > Authors

---

> > > > > ### Author Response · Authors · 2023-11-22
> > > > > **Keenly Seeking Your Feedback on Revised Manuscript**
> > > > >
> > > > > Dear Reviewer 2KBE,
> > > > >
> > > > > We sincerely appreciate your insightful comments on our manuscript. Your valuable suggestions have been incorporated into our revision, and we are eager to receive your feedback.
> > > > >
> > > > > We are particularly encouraged by your comment
> > > > >
> > > > > > "I am looking forward to the authors’ response."
> > > > >
> > > > > With the author-reviewer discussion phase concluding shortly, we would like to check if there are any lingering concerns or areas that may require further clarification. Your support in this final phase, especially if you find the revisions satisfactory, would be of immense significance. Your feedback and evaluation play a pivotal role in determining the ultimate fate of our work.
> > > > >
> > > > > regards,
> > > > >
> > > > > Authors

---

> > > > > > ### Author Response · Authors · 2023-11-23
> > > > > > **Earnestly awaiting feedback from Reviewer 2KBE**
> > > > > >
> > > > > > Dear Reviewer 2KBE,
> > > > > >
> > > > > > We earnestly appeal to you to share your feedback on our revised version. We are happy to inform you that all of the three reviewers now lean towards acceptance. Your insights and evaluation play a crucial role in deciding the ultimate fate of our work, and we are eagerly awaiting your response to the revised manuscript.
> > > > > >
> > > > > > regards,
> > > > > >
> > > > > > Authors.

---

> > > > > > > ### Comment · Reviewer_2KBE · 2023-11-23
> > > > > > > **Thanks for the rebuttal**
> > > > > > >
> > > > > > > Thanks for the response. I will increase my score to 6.

---

### Public Comment · ~Jiahao_Wu3 · 2023-11-14

The approach presented in this paper can be characterized more as a sub-structure mining algorithm than a graph distillation method. Nevertheless, it introduces a novel perspective on graph distillation. Empirical results demonstrate comparable performance to traditional graph distillation, a noteworthy observation. While the condensation rate and classification performance show promise, it is advisable for the authors to provide a clear rationale for the significantly smaller sizes of computation tree sets compared to the sizes of graphs. This explanation is crucial, as the process of decomposing a graph into a set of computation trees might intuitively be expected to result in larger storage requirements due to potential overlap of nodes in non-isometric trees.

---

> ### Author Response · Authors · 2023-11-19
>
> To explain the high compression achieved by MIRAGE, we first would like to point to Figure 1 in the paper and the discussion associated with it in Section 3.2. We observe that frequency distribution of computation trees is a long-tail distribution. The plot shows that a compact set of top-k frequent trees(non-isometric) captures a substantial portion of the distribution mass of the dataset. As the distillation process of Mirage builds the condensed dataset using **only** these frequent trees, hence, the storage is low. Furthermore, these trees are only n layers deep, where n is the number of layers in GNN. Typically n is within 3.
>
> Finally, we also point to the sufficiency experiments in Sec 4.4. The experiments indicate that the loss is similar for both the distilled dataset and the full dataset when using a model with weights trained on the full dataset. Thus, showcasing the GNN is attending to these frequent patterns primarily. This behavior by the GNN is not surprising. Specifically, gradient descent will run more iterations on trees that it sees more often and hence infrequent trees have limited impact on the gradients.

---

> > ### Public Comment · ~Jiahao_Wu3 · 2023-11-24
> >
> > Thanks for your detailed explanation.

---

### Author Response · Authors · 2023-11-18
**Summary of rebuttal**

We thank the reviewers for their insights and constructive suggestions. A comprehensive point-by-point response to the reviewers' comments is presented below. We have updated the main manuscript and the appendix to address these comments. The changes made in the manuscript are highlighted in *blue* color. The *major additional changes* are listed below.

* **Additional experiments:** We have incorporated all of the additional experiments requested by the reviewers spanning
    - extensive parameter sensitivity analysis
    - empirical validation of information content in frequent patterns
    - generalization to node classification
    - multi-class classification

* **Presentation**: We have addressed several presentation-related inquiries, such as better illustration of the frequency distribution of computation trees (Fig. 1) and the usage of the "unsupervised" term in describing MIRAGE.

We hope these revisions will satisfactorily address the concerns raised by the reviewers and elevate the overall quality of our work.

---

### Author Response · Authors · 2023-11-20
**Looking forward to your feedback on rebuttal**

Dear Reviewers,

Thank you once again for all of your constructive comments, which have helped us significantly improve the paper! As detailed below, we have performed several additional experiments and analyses to address the comments and concerns raised by the reviewers.

Since we are into the last two days of the discussion phase, we are eagerly looking forward to your post-rebuttal responses.

Please do let us know if there are any additional clarifications or experiments that we can offer. We would love to discuss more if any concern still remains. Otherwise, we would appreciate it if you could support the paper by increasing the score.

Thank you!

Authors

---

### Meta-Review · Area_Chair_A8ma · 2023-12-05

**Metareview:**

An interesting contribution to graph distillation, MIRAGE, is proposed here. Leveraging the computational tree representations induced by message passing models, the authors construct appropriate compressions which demonstrate convincing performance across a variety of datasets. After a successful rebuttal, the reviewers are unanimous in recommending acceptance. I have no reservations to back this recommendation.

**Justification For Why Not Higher Score:**

I find that in its present form the paper just about exceeds the bar of ICLR publication---no reviewer championed it for a stronger outcome. I also think that a more comprehensive comparison over types of GNN baselines (including models like MPNN) would have been paramount for this work to make it to the next level.

**Justification For Why Not Lower Score:**

The reviewers unanimously recommend acceptance after the Authors' rebuttal, and I am in agreement with them.

---

### Decision · Program_Chairs · 2024-01-16

Accept (poster)